# Male-biased Cyp17a2 orchestrates antiviral sexual dimorphism in fish via STING stabilization and viral protein degradation

Long-Feng Lu[1,2†], Bao-jie Cui[1,3†], Sheng-Chi Shi[1,2†], Yang-Yang Wang[1,2], Can Zhang[1,2], Xiao Xu[1,3], Meng-Ze Tian[1,2], Zhen-Qi Li[1,2], Na Xu[1,2], Zhuo-Cong Li[1,2], Dan-Dan Chen[1,2], Li Zhou[1,2], Gang Zhai[1,2*], Zhan Yin[2,4], Shun Li[1,2,3,5,6*]

[1]Key Laboratory of Breeding Biotechnology and Sustainable Aquaculture, Institute of Hydrobiology, Chinese Academy of Sciences, Wuhan, China; [2]University of Chinese Academy of Sciences, Beijing, China; [3]College of Fisheries and Life Science, Dalian Ocean University, Dalian, China; [4]Key Laboratory of Mariculture, Ministry of Education, Ocean University of China, Qingdao, China; [5]Laboratory for Marine Biology and Biotechnology, Qingdao Marine Science and Technology Center, Qingdao, China; [6]Key Laboratory of Aquaculture Disease Control, Ministry of Agriculture, Wuhan, China

*For correspondence:
zhaigang@ihb.ac.cn (GZ);
bob@ihb.ac.cn (SL)

†These authors contributed equally to this work

## eLife Assessment

This **valuable** study describes an interesting infection phenotype that differs between adult male and female zebrafish. The authors present data indicating that male-biased expression of Cyp17a2 appears to mediate viral infection through STING and USP8 activity regulation. Through experimentation on male fish, the authors present **solid** evidence linking this factor to direct and indirect antiviral outcomes through ubiquitination pathways. These findings raise interesting questions about immune mechanisms that underlie sex-dimorphism and the selective pressures that might shape it.

**Abstract** Differences in immunity between males and females in living organisms are generally thought to be due to sex hormones and sex chromosomes, and it is often assumed that males have a weaker immune response. Here, we report that in fish, males exhibit stronger antiviral immune responses, the male-biased gene *cyp17a2* as a critical mediator of this enhanced response. First, we observed that male zebrafish exhibit enhanced antiviral resistance compared to females, and notably, zebrafish lack sex chromosomes. Through transcriptomic screening, we found that *cyp17a2* was specifically highly expressed in male fish. *Cyp17a2* knockout males were equivalent to wild-type males in terms of sex organs and androgen secretion, but the ability to upregulate IFN as well as antiviral resistance was greatly reduced. Then, Cyp17a2 is identified as a positive IFN regulator which is located at the endoplasmic reticulum and specifically interacts with and enhances STING-mediated antiviral responses. Mechanistically, Cyp17a2 stabilizes STING expression by recruiting the E3 ubiquitin ligase bloodthirsty-related gene family member 32 (btr32), which facilitates K33-linked polyubiquitination. The capacity of IFN induction of Cyp17a2 was abolished when STING was knocked down. Meanwhile, Cyp17a2 also attenuates viral infection directly to strengthen the antiviral capacity. As an antiviral protein, Cyp17a2 degrades the spring viremia of carp virus (SVCV) P protein by utilizing USP8 to reduce its K33-linked polyubiquitination. These findings reveal a sex-based regulatory mechanism in teleost antiviral immunity, broadening our understanding of

sexual dimorphism in immune responses beyond the conventional roles of sex chromosomes and hormones.

## Introduction

Research has demonstrated that innate immune responses to viruses differ significantly between the sexes. Males exhibit heightened susceptibility to viral pathogens, correlating with their generally weaker innate immune activation compared to females (*Klein and Huber, 2010*). This immunological dimorphism manifests clinically as elevated infection intensity (quantified by intra-host viral load) and prevalence (population-level infection rates) in males (*Klein, 2012*). Notably, sex-specific clinical outcomes are exemplified by severe acute respiratory syndrome coronavirus 2 (SARS-CoV-2) infections, where epidemiological data suggest males face twice the risk of intensive care unit (ICU) admission and a 30% higher mortality rate compared to females (*Health et al., 2020*). Similar trends extend to other viral pathogens including Dengue virus, hantaviruses, and hepatitis B/C viruses, all exhibiting male-biased infection prevalence (*Guha-Sapir and Schimmer, 2005*; *Tsay et al., 2009*; *Burguete-García et al., 2011*).

Several factors have been identified as contributors to these sex-specific disparities in antiviral immunity, including hormonal variations and sex chromosomes. For instance, androgens such as testosterone exhibit broad immunosuppressive effects, including reduced NK cell cytotoxicity, impaired neutrophil chemotaxis, and attenuated macrophage proinflammatory cytokine production in vitro (*Liva and Voskuhl, 2001*). In addition, the Y chromosome harbors only two annotated miRNAs, while the X chromosome contains numerous immune-relevant miRNAs, and emerging evidence implicates X-escapee miRNAs as key modulators of sex-biased immune responses (*Pinheiro et al., 2011*; *Sharma and Eghbali, 2014*).

It is well known that the interferon (IFN) response is an important host immune response to viral infection, in both mammals and fish (*Akira et al., 2006*). The signaling pathways that activate IFN expression, although slightly different in fish and mammals, are generally conserved in both (*Chen et al., 2017*). The endoplasmic reticulum (ER) protein stimulator of IFN genes (STING) is an important antiviral protein capable of resisting both DNA and RNA viruses by activating TANK binding kinase 1 (TBK1) during the induction of IFN production, which in turn phosphorylates IFN regulatory factor 3 (IRF3) in the nucleus and activates the signaling of IFN transcription (*Ishikawa and Barber, 2008*; *Ni et al., 2018*). For instance, STING deficiency has been shown to impair innate immune responses against multiple RNA viruses, such as dengue virus, West Nile virus, and Japanese encephalitis virus (*Aguirre et al., 2012*; *Nazmi et al., 2012*; *You et al., 2013*). Notably, oncogenes of the DNA tumor viruses, including human papillomavirus E7 and adenovirus E1A, function as potent and specific inhibitors of the cyclic GMP-AMP synthase (cGAS)-STING signaling pathway (*Lau et al., 2015*).

Cyp17a is classified within the cytochrome P450 (CYP450) family. In mammals, there is only one form, cyp17a1, which possesses both hydroxylase and lyase functions. Mutations in the human *cyp17a1* gene are associated with congenital adrenal hyperplasia (CAH) (*Kim et al., 2014*). Comparative genomic analyses reveal that teleost fishes possess two distinct *cyp17a* isoforms, the evolutionarily conserved *cyp17a1* and a phylogenetically divergent *cyp17a2* paralog, with the latter representing a teleost-specific innovation absent in mammalian genomes (*Zhou et al., 2007*). Current investigations of Cyp17a2 in teleost species have predominantly centered on its mechanistic roles governing sexual development and differentiation, for example, with knockout of zebrafish *cyp17a2* leading to significant impairment of sperm motility (*Shi et al., 2022*). Although Cyp17a2 has been characterized in sexual development, its non-canonical physiological roles have yet to be fully elucidated.

In most vertebrates, the XY sex chromosomes exhibit significant physiological differences between males and females, as exemplified in humans. However, the sex determination mechanisms in fish are more complex. For instance, zebrafish lack sex chromosomes, which confers a natural advantage. This absence allows for the exclusion of sex chromosome-related differences in the study of immunological disparities between males and females. Consequently, we selected zebrafish as the model organism for this investigation. This enabled us to more precisely identify the genes that are differentially expressed in male and female autosomes, which is crucial for understanding the immunological differences between the sexes. This study elucidates the role of the teleost-specific gene *cyp17a2* in antiviral immunity. Furthermore, the discovery that the male-biased expression of *cyp17a2* confers

**eLife digest** Sex differences can influence physiological responses to exercise, stress or immune responses. For example, males are often more susceptible to viral infections due to hormonal and chromosomal influences.

While primary sex determination in mammals is strictly linked to chromosomes, other animals, such as fish, can have a very fluid sex determination, with some fish species lacking sex chromosomes completely. This diversity provides a unique system to study how sexual dimorphism affects immune responses, independent of classical sex-linked pathways.

Long-Feng et al. used a fish model with alternative sex determination to test whether non-sex-chromosome genes can affect immune differences between males and females, without the influence of sex chromosomes or hormones. Using genetic and biochemical methods, the researchers demonstrated that male zebrafish are more resistant than females to virus, despite the absence of sex chromosomes. They identified an enzyme called Cyp17a2 to be responsible for this difference, which was present in high amounts in males.

This enzyme is usually involved in steroid hormone production, but in species such as zebrafish, it regulates stress hormone production. The experiments revealed that Cyp17a2 boosts antiviral defense in males in two ways. First, it strengthens the fish's immune signaling by stabilizing a protein known as STING. It does this by adding ubiquitin, a chemical tag that helps the protein to work properly. Second, it directly suppresses viral replication by promoting the breakdown of a viral protein. Moreover, male fish genetically engineered to lack this protein showed reduced immune responses. Overall, these findings suggest that Cyp17a2 is an immune regulator that connects male-specific traits to antiviral defenses.

By studying a vertebrate model with alternative sex determination, Long-Feng et al. challenged the mammal-centric view of immune sexual dimorphism that centers on sex chromosomes and hormones and provided new insight into how sex-specific immunity may have evolved. The results prompt reconsideration of the unique mechanisms underlying sex-based immune differences in fish. They may also inform the development of sex-specific strategies to improve disease resistance in economically important fish. Future work should assess the evolutionary conservation of this pathway across vertebrates and explore whether targeting the Cyp17a2 and STING could offer new approaches to antiviral therapy, although clinical applications remain speculative.

enhanced resistance to infection in males provides valuable insight into the complex mechanisms underlying sexual immune dimorphism.

## Results

### Male fish exhibit enhanced resistance to viral infection compared to female fish

To determine whether sex differences exist in the antiviral response capacity of fish, age-matched female and male zebrafish were intraperitoneally (i.p.) injected with SVCV. Male fish exhibited a significantly higher survival rate than female fish within 7 days post-infection (*Figure 1A*). Concurrently, females demonstrated a markedly higher incidence of cutaneous hemorrhages relative to males (*Figure 1B*). Consistent with the reduced mortality, male fish showed less tissue damage in the liver and spleen following viral infection (*Figure 1C*). IF analysis revealed that the SVCV N protein, represented by green fluorescence, was barely detectable in male tissues but was prominently present in female tissues (*Figure 1D*). At the mRNA level, viral transcripts were quantified in these tissues, and the abundance of *svcv-n* gene was significantly lower in male tissues (*Figure 1E*). At the protein level, SVCV N and G proteins were less frequently detected in male tissues compared to female tissues in seven samples (*Figure 1F*). Additionally, viral titers were significantly lower in the hearts of male fish compared to female fish (*Figure 1G*). These findings demonstrate a stronger antiviral response capacity in male fish compared to female fish.

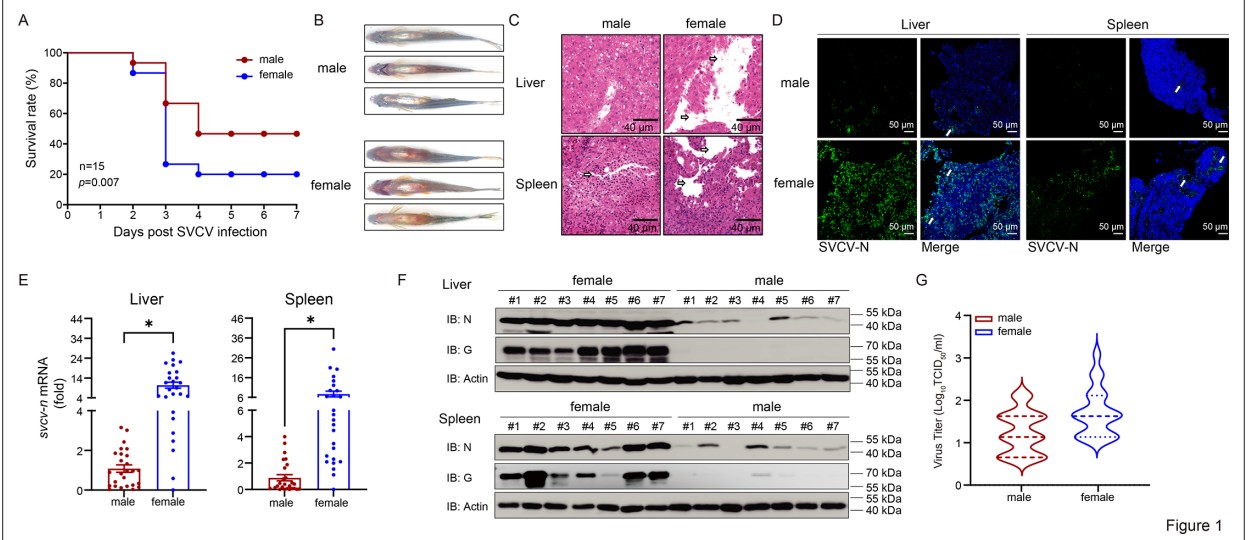

**Figure 1.** Male fish show stronger viral resistance than females. (**A**) Survival (Kaplan-Meier Curve) of male and female zebrafish (n=15 per group) at various days after i.p. injected with SVCV ($5×10^8$ $TCID_{50}$/ml, 5 μl/individual). (**B**) Sex-specific morphological alterations in zebrafish given i.p. injection of SVCV for 48 hr (n=3 per group). (**C**) Microscopy of H&E-stained liver and spleen sections from male and female zebrafish treated with SVCV for 48 hr. (**D**) IF analysis of SVCV-N protein in liver and spleen of male and female zebrafish treated with SVCV for 48 h. (**E**) qPCR analysis of *svcv-n* mRNA in the liver and spleen of male and female zebrafish (n=26 per group) given i.p. injection of SVCV for 48 hr. (**F**) IB analysis of SVCV proteins in the liver and spleen sections of male and female zebrafish (n=7 per group) treated with SVCV for 48 hr. (**G**) The viral titer of heart in male and female zebrafish (n=40 per group) treated with SVCV for 48 hr.

The online version of this article includes the following source data for figure 1:

**Source data 1.** PDF file containing original western blots for *Figure 1F* indicating the relevant bands and treatments.

**Source data 2.** Original files for western blot analysis displayed in *Figure 1F*.

**Source data 3.** Original data for graphs analysis in *Figure 1A, E and G*.

## Male-biased gene cyp17a2 orchestrates antiviral immunity through RLR pathway

To explore the potential causes of the observed differences in virus resistance between male and female fish, transcriptome sequencing was performed on head-kidney tissues of healthy adult male and female zebrafish. Differential expression analysis identified 1511 upregulated genes and 1117 down-regulated genes (*Figure 2A* and *Supplementary file 2*). From these, we focused on a subset of known or putative sex-related genes. Among eight sex-related genes, *cyp17a2* exhibited the most significant male-biased upregulation, which was subsequently confirmed by qPCR (*Figure 2B*, *Figure 2—figure supplement 1A*). Furthermore, we systematically quantified Cyp17a2 mRNA and protein expression in multiple tissues of male and female zebrafish, demonstrating a consistent male-biased expression pattern of Cyp17a2 (*Figure 2C and D*). Thus, Cyp17a2 was selected for further study. We first sought to understand whether it was associated with host antiviral immunity, and the significant upregulation of the Cyp17a2 at the mRNA and protein level in response to viral stimulation suggested its possible involvement in the antiviral process (*Figure 2—figure supplement 1B and C*). Following the generation of *cyp17a2* knockout zebrafish, a previous study revealed that male *cyp17a2*$^{-/-}$ mutants displayed normal development of male-typical secondary sexual characteristics comparable to WT males. However, female *cyp17a2*$^{-/-}$ mutants exhibited significantly underdeveloped genital papillae compared to their WT counterparts. To eliminate potential confounding effects of androgen-mediated pathways on immune parameters, subsequent experiments were conducted exclusively using male specimens from both genotypes. The male *cyp17a2*$^{-/-}$ mutants exhibited a higher mortality rate than WT males after viral infection (*Figure 2E*). Significant tissue damage and strong green fluorescence representing the SVCV N protein was observed in the *cyp17a2*$^{-/-}$ tissues. Virus titers assay also confirmed that viral proliferation was higher in the *cyp17a2*$^{-/-}$ group (*Figure 2—figure supplement 1D–F*). Consistent with these observations, the replication of viral genes transcription and protein levels was increased

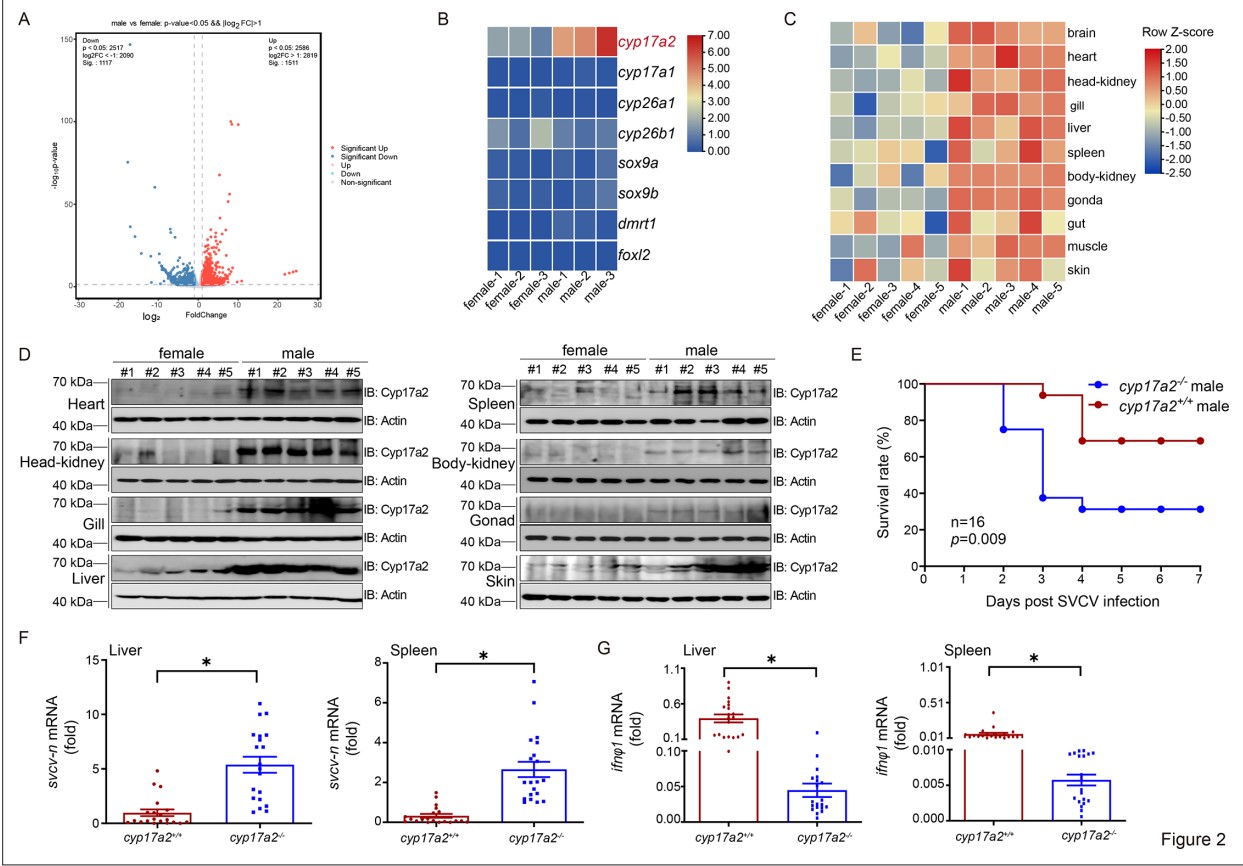

**Figure 2.** Male-biased cyp17a2 regulates RLR-mediated antiviral immunity. (**A**) The differently expressed gene number of mRNA variations presented as a volcano plot in head-kidney of male and female zebrafish. (**B**) Heatmap view of mRNA variations of eight sex-regulated genes in the head-kidney of male and female zebrafish. Values are represented as $\log_2^{(FPKM+1)}$. (**C**) Heatmap view of *cyp17a2* mRNA in the brain, heart, head-kidney, gill, liver, spleen, body-kidney, gonad, gut, muscle, and skin of male and female zebrafish (n=5 per group). Expression values [$\log_2^{(fold\ change)}$] were transformed to Z-scores across rows for each gene to highlight patterns of relative expression. (**D**) IB analysis of Cyp17a2 in the heart, head-kidney, gill, liver, spleen, body-kidney, gonad, and skin of male and female zebrafish (n=5 per group). (**E**) Survival (Kaplan-Meier Curve) of *cyp17a2$^{+/+}$* and *cyp17a2$^{-/-}$* male zebrafish (n=16 per group) at various days after i.p. injected with SVCV ($5\times10^8$ TCID50/ml, 5 μl/individual). (**F**) qPCR analysis of *svcv-n* mRNA in the liver and spleen of *cyp17a2$^{+/+}$* and *cyp17a2$^{-/-}$* male zebrafish (n=20 per group) given i.p. injection of SVCV for 48 hr. (**G**) qPCR analysis of *ifnφ1* mRNA in the liver and spleen of *cyp17a2$^{+/+}$* and *cyp17a2$^{-/-}$* male zebrafish (n=20 per group) given i.p. injection of SVCV for 48 hr.

The online version of this article includes the following source data and figure supplement(s) for figure 2:

**Source data 1.** PDF file containing original western blots for *Figure 2D* indicating the relevant bands and treatments.

**Source data 2.** Original files for western blot analysis displayed in *Figure 2D*.

**Source data 3.** Original data for graphs analysis in *Figure 2B, C and E–G*.

**Figure supplement 1.** The male-biased expression of Cyp17a2 governs RLR antiviral signaling.

**Figure supplement 1—source data 1.** PDF file containing original western blots for *Figure 2—figure supplement 1C and G* indicating the relevant bands and treatments.

**Figure supplement 1—source data 2.** Original files for western blot analysis displayed in *Figure 2—figure supplement 1C and G*.

**Figure supplement 1—source data 3.** Original data for graphs analysis in *Figure 2—figure supplement 1A, B and F*.

**Figure supplement 2.** Heatmap view of mRNA variations of RLR-mediated ISG sets in the liver of *cyp17a2$^{+/+}$* and *cyp17a2$^{-/-}$* male zebrafish infected with SVCV.

in *cyp17a2$^{-/-}$* homozygote tissues. These data demonstrated that Cyp17a2 is crucial for host antiviral defense (*Figure 2F*, *Figure 2—figure supplement 1G*). To further understand their functional mechanisms, a transcriptomic analysis contains a total of 3114 differentially expressed genes (DEGs) were identified, there are 1217 genes upregulated and 1897 were downregulated in the liver of *cyp17a2* knockout male zebrafish compared to WT males after viral infection (*Figure 2—figure supplement*

*1H*). Kyoto Encyclopedia of Genes and Genomes (KEGG) pathway analysis revealed RIG-I-like receptors (RLRs), NOD-like receptors (NLRs), and Toll-like receptors (TLRs) signaling pathways were significantly affected (*Figure 2—figure supplement 1I*). Gene set enrichment analysis (GSEA) showed that RLRs were downregulated in the *cyp17a2⁻/⁻* group under SVCV infection, combining with numerous IFN-stimulated genes (ISGs) decreases (*Figure 2—figure supplements 1J and 2*). To validate the transcriptome data, *ifnφ1* levels were assessed and identified as significantly downregulated in the *cyp17a2⁻/⁻* group (*Figure 2G*). Collectively, these data suggest that viral proliferation was enhanced and IFN response was reduced in *cyp17a2⁻/⁻* fish.

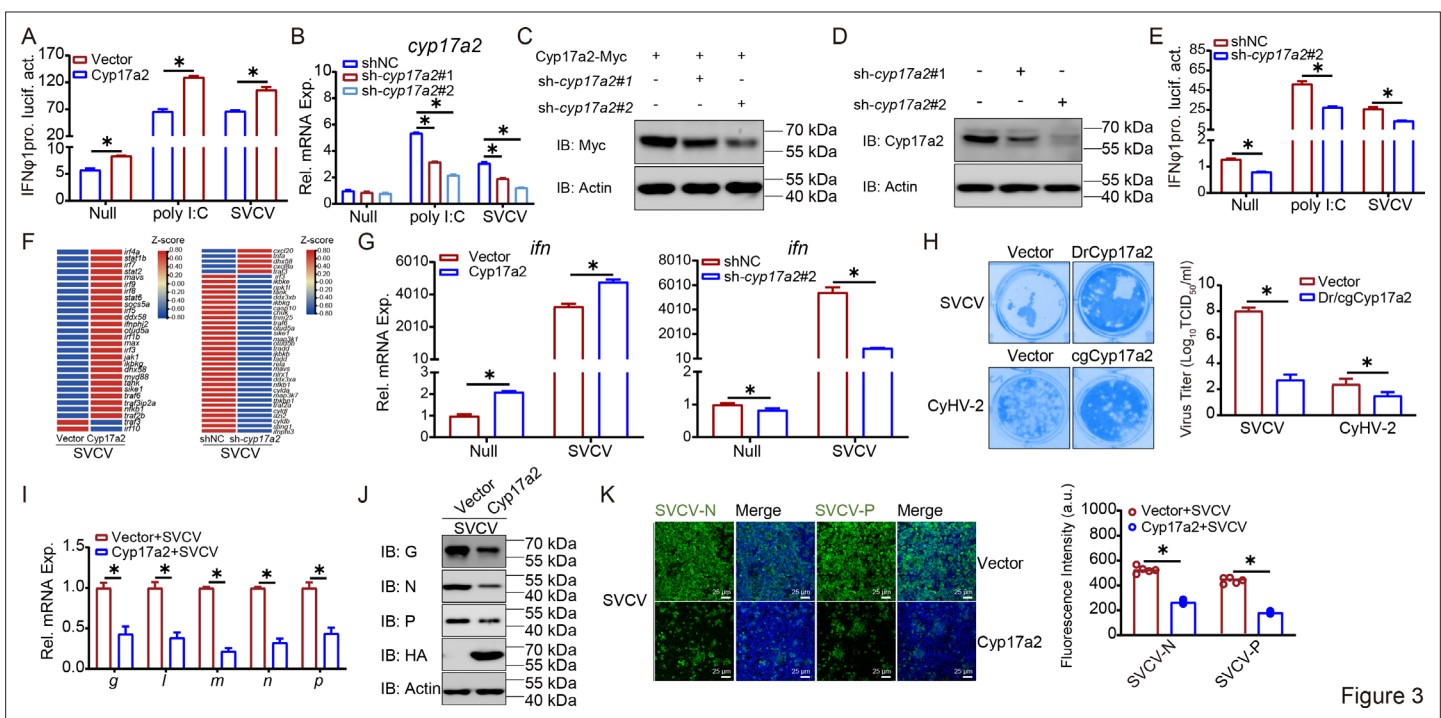

**Figure 3.** Cyp17a2 upregulates IFN expression and inhibits viral replication. (**A and E**) Luciferase activity of IFNφ1pro in EPC cells transfected with indicated plasmids for 24 hr, and then untreated or transfected with poly I:C (0.5 µg) or infected with SVCV (MOI = 1) for 24 hr before luciferase assays. (**B**) qPCR analysis of *cyp17a2* in EPC cells transfected with indicated plasmids for 24 hr, and then untreated or transfected with poly I:C (2 µg) or infected with SVCV (MOI = 1) for 24 hr. (**C and D**) IB analysis of proteins in EPC cells transfected with indicated plasmids for 24 hr. (**A–E**) Representative experiments are shown (n=3). (**F**) Heatmap view of mRNA variations of SVCV-activated ISG sets in the Cyp17a2-overexpressing cells or *cyp17a2* knockdown cells and infected with SVCV for 24 hr. Expression values are log₂^(FPKM+1) transformed and presented as row Z-scores. (**G**) qPCR analysis of *ifn* in EPC cells transfected with indicated plasmids for 24 hr, and then untreated or infected with SVCV (MOI = 1) for 24 hr. (**H**) Plaque assay of virus titers in EPC cells or GICB cells transfected with indicated plasmids for 24 hr, followed by SVCV or CyHV-2 challenge for 24 hr or 48 hr. (**I and J**) qPCR and IB analysis of SVCV genes in EPC cells transfected with indicated plasmids for 24 hr, followed by SVCV challenge for 24 hr. (**K**) IF analysis of SVCV proteins in EPC cells transfected with indicated plasmids for 24 hr, followed by SVCV challenge for 24 hr. The fluorescence intensity (arbitrary unit, a.u.) was recorded by the LAS X software, and the data were expressed as mean ± SD, n=5. (**G–K**) Representative experiments are shown (n=3).

The online version of this article includes the following source data and figure supplement(s) for figure 3:

**Source data 1.** PDF file containing original western blots for *Figure 3C, D and J* indicating the relevant bands and treatments.

**Source data 2.** Original files for western blot analysis displayed in *Figure 3C, D and J*.

**Source data 3.** Original data for graphs analysis in *Figure 3A, B, E, G–I and K*.

**Figure supplement 1.** Cyp17a2 enhances IFN production to inhibit viral replication.

**Figure supplement 1—source data 1.** PDF file containing original western blots for *Figure 3—figure supplement 1H* indicating the relevant bands and treatments.

**Figure supplement 1—source data 2.** Original files for western blot analysis displayed in *Figure 3—figure supplement 1H*.

**Figure supplement 1—source data 3.** Original data for graphs analysis in *Figure 3—figure supplement 1A-G and I*.

## Cyp17a2 enhances IFN expression and suppresses viral proliferation

Given the critical role of the IFN response in host defense against viral infections, we investigated the impact of Cyp17a2 on IFN expression and viral infection in vitro. Overexpression of Cyp17a2 significantly upregulated the IFN promoter and ISRE activities in response to poly I:C or SVCV stimulation (*Figure 3A*, *Figure 3—figure supplement 1A-B*). An effective sh-*cyp17a2*#2 was generated and confirmed at both the mRNA and protein levels, and knockdown of *cyp17a2* inhibited both the IFN promoter and ISRE activity (*Figure 3B-E*, *Figure 3—figure supplement 1C-D*). RNA sequencing analysis of the cell line revealed that SVCV infection activated numerous ISGs in Cyp17a2-overexpressing cells, whereas these ISGs were significantly suppressed upon *cyp17a2* knockdown (*Figure 3F* and *Supplementary file 3*). The induction or attenuation of IFN and ISGs upon Cyp17a2 overexpression or knockdown was confirmed by qPCR analysis, respectively (*Figure 3G*, *Figure 3—figure supplement 1E*). For antiviral capacity assays, Cyp17a2 from zebrafish and gibel carp both inhibited the proliferation of host-specific viruses; conversely, *cyp17a2* knockdown significantly enhanced viral proliferation (*Figure 3H*, *Figure 3—figure supplement 1F*). Regarding viral mRNA and proteins, overexpression or knockdown of Cyp17a2 also displayed similar opposite findings (*Figure 3I-J*, *Figure 3—figure supplement 1G-H*). IF assays indicated a lower intensity of green signals representing the viral protein in the Cyp17a2 overexpression group compared to the control, while a higher green signal was observed in the *cyp17a2* knockdown group relative to the normal group (*Figure 3K*, *Figure 3—figure*

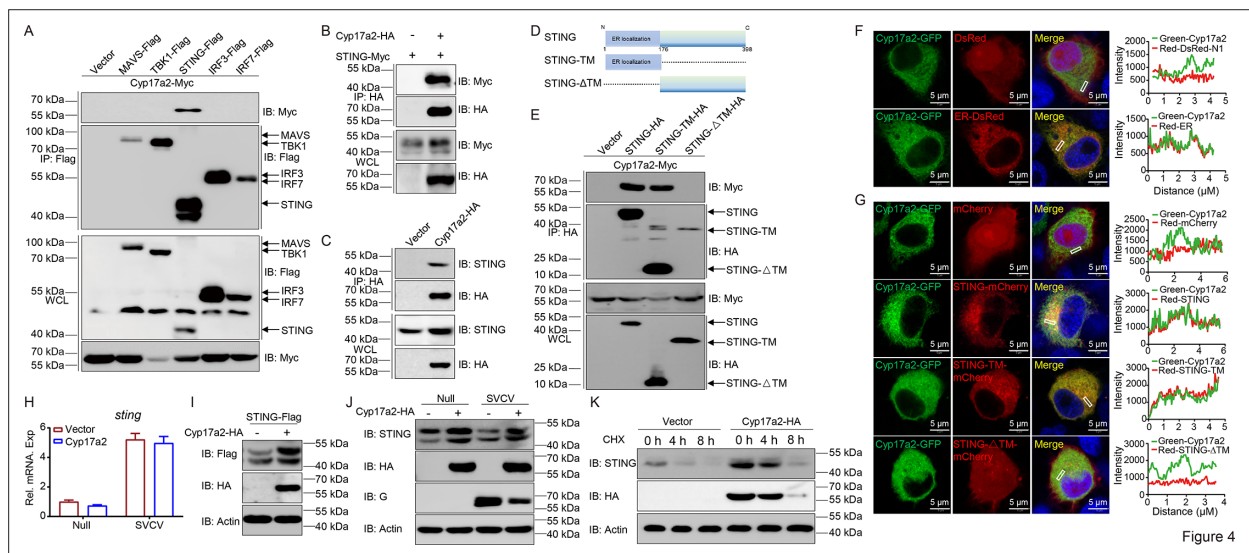

**Figure 4.** Cyp17a2 physically interacts with STING and stabilizes its expression. (**A–C and E**) IB analysis of WCLs and proteins immunoprecipitated with anti-Flag or HA Ab-conjugated agarose beads from EPC cells transfected with indicated plasmids for 24 hr. Representative experiments are shown (n=3). (**D**) Schematic representation of full-length STING and its mutants. (**F and G**) Confocal microscopy of Cyp17a2 and ER or STING and its mutants in EPC cells transfected with indicated plasmids for 24 hr. The coefficient of colocalization was determined by qualitative analysis of the fluorescence intensity of the selected area in Merge. (**H**) qPCR analysis of *sting* in EPC cells transfected with indicated plasmids for 24 hr, followed by untreated or infected with SVCV (MOI = 1) for 24 hr. (**I**) IB analysis of proteins in EPC cells transfected with indicated plasmids for 24 hr. (**J**) IB analysis of proteins in EPC cells transfected with indicated plasmids for 24 hr, followed by untreated or infected with SVCV (MOI = 1) for 24 hr. (**K**) IB analysis of proteins in EPC cells transfected with indicated plasmids for 18 hr, then treated with CHX for 4 hr and 8 hr. (**F–K**) Representative experiments are shown (n=3).

The online version of this article includes the following source data and figure supplement(s) for figure 4:

**Source data 1.** PDF file containing original western blots for *Figure 4A-C, E and I–K* indicating the relevant bands and treatments.

**Source data 2.** Original files for western blot analysis displayed in *Figure 4A–C, E and I–K*.

**Source data 3.** Original data for graphs analysis in *Figure 4F, G and H*.

**Figure supplement 1.** STING stability is enhanced by Cyp17a2.

**Figure supplement 1—source data 1.** PDF file containing original western blots for *Figure 4—figure supplement 1C–E* indicating the relevant bands and treatments.

**Figure supplement 1—source data 2.** Original files for western blot analysis displayed in *Figure 4—figure supplement 1C–E*.

**Figure supplement 1—source data 3.** Original data for graphs analysis in *Figure 4—figure supplement 1A and B*.

*supplement 1I*). These results suggest that Cyp17a2 upregulates IFN expression and enhances antiviral capacity in the host.

## Cyp17a2 interacts with STING and stabilizes its expression

Given the critical role of the RLR signaling pathway in fish IFN activation, the relationship between Cyp17a2 and RLR signaling adaptors was investigated. Co-transfection with Myc-Cyp17a2 and Flag-tagged RLR components (MAVS, TBK1, STING, IRF3, IRF7) followed by anti-Flag immunoprecipitation identified specific Cyp17a2-STING interaction via anti-Myc detection (*Figure 4A*). Reciprocal Co-IP and semi-endogenous IP assays confirmed this interaction (*Figure 4B and C*). Truncated STING mutants were generated to map binding domains (*Figure 4D*). The transmembrane domain (TM)-deficient STING-ΔTM-HA failed to bind Cyp17a2, indicating TM domain dependence (*Figure 4E*). Cyp17a2-GFP overlapped with ER markers in confocal microscopy and colocalized with STING but failed to colocalize with the ΔTM mutant (*Figure 4F-G*, *Figure 4—figure supplement 1A*). To identify the effect of Cyp17a2 on STING regulation, the mRNA level of sting was first analyzed; minimal change was observed in *sting* upon Cyp17a2 overexpression (*Figure 4H*). Protein-level studies demonstrated Cyp17a2 remarkably enhanced STING abundance, confirmed by confocal microscopy (*Figure 4I*, *Figure 4—figure supplement 1B*). Endogenous STING was similarly stabilized dependent on Cyp17a2 under untreated or stimulated conditions (*Figure 4J*, *Figure 4—figure supplement 1C*). Cycloheximide (CHX) chase assay showed prolonged exogenous and endogenous STING half-life with Cyp17a2 co-expression, and it was also confirmed by knockdown experiments (*Figure 4K*, *Figure 4—figure supplement 1D-E*). These data demonstrate that Cyp17a2 binds STING via its TM and enhances STING posttranslational stability.

## Cyp17a2 enhances STING-mediated IFN expression and antiviral responses

Considering that Cyp17a2 targets STING, a positive regulator of IFN production, we investigated whether Cyp17a2 influences the IFN expression induced by STING. The IFNφ1pro activity induced by STING overexpression was significantly enhanced by Cyp17a2, with similar amplification observed for ISRE promoter activation (*Figure 5A*, *Figure 5—figure supplement 1A*). Conversely, *cyp17a2* knockdown was shown to impair STING-mediated IFNφ1 pro and ISRE activation (*Figure 5B*, *Figure 5—figure supplement 1B*). Corresponding increases in *ifn* and *vig1* transcripts were detected in Cyp17a2 overexpressing cells, while Cyp17a2 knockdown suppressed these responses (*Figure 5C-D*, *Figure 5—figure supplement 1C-D*). The subsequent investigation focused on the potential modulatory effect of Cyp17a2 on the STING-induced cellular antiviral response. STING overexpression reduced cytopathogenic effect (CPE), with viral titers decreased 336-fold compared to the control group. This protection was augmented by Cyp17a2 co-expression, achieving an additional 20-fold reduction (*Figure 5E*, *Figure 5—figure supplement 1E*). IB and IF analysis confirmed that Cyp17a2 potentiated STING-mediated suppression of viral protein expression (*Figure 5F*, *Figure 5—figure supplement 1F*). SVCV gene transcripts were suppressed by STING, with enhanced inhibition upon Cyp17a2 co-expression (*Figure 5G*). STING dependency was validated using shRNA knockdown (*Figure 5—figure supplement 1G*). The antiviral capacity enhanced by Cyp17a2 was markedly attenuated in cells with *sting* knockdown, as evidenced by restored viral titers, gene and protein expression (*Figure 5H-J*, *Figure 5—figure supplement 1H-I*). These results demonstrate that Cyp17a2 potentiates STING activity to enhance IFN production and antiviral responses.

## Cyp17a2 stabilizes STING through btr32-mediated K33-linked polyubiquitination

To elucidate the molecular mechanism by which Cyp17a2 augments STING stabilization, proteasomal and lysosomal degradation were pharmacologically inhibited using MG132 and Chloroquine (CQ), respectively. Accelerated STING degradation caused by *cyp17a2* knockdown was rescued by MG132 but not CQ, suggesting knockdown of *cyp17a2* promoted STING degradation via the proteasome pathway (*Figure 6A*). Then, enhanced polyubiquitination of STING was detected upon Cyp17a2 co-expression (*Figure 6B*). Ubiquitin linkage analysis revealed selective enhancement of K33-linked chains, with *cyp17a2* knockdown shown to reduce both WT and K33-specific ubiquitination (*Figure 6C*, *Figure 6—figure supplement 1A*). Subsequently, candidate E3 ligases were screened,

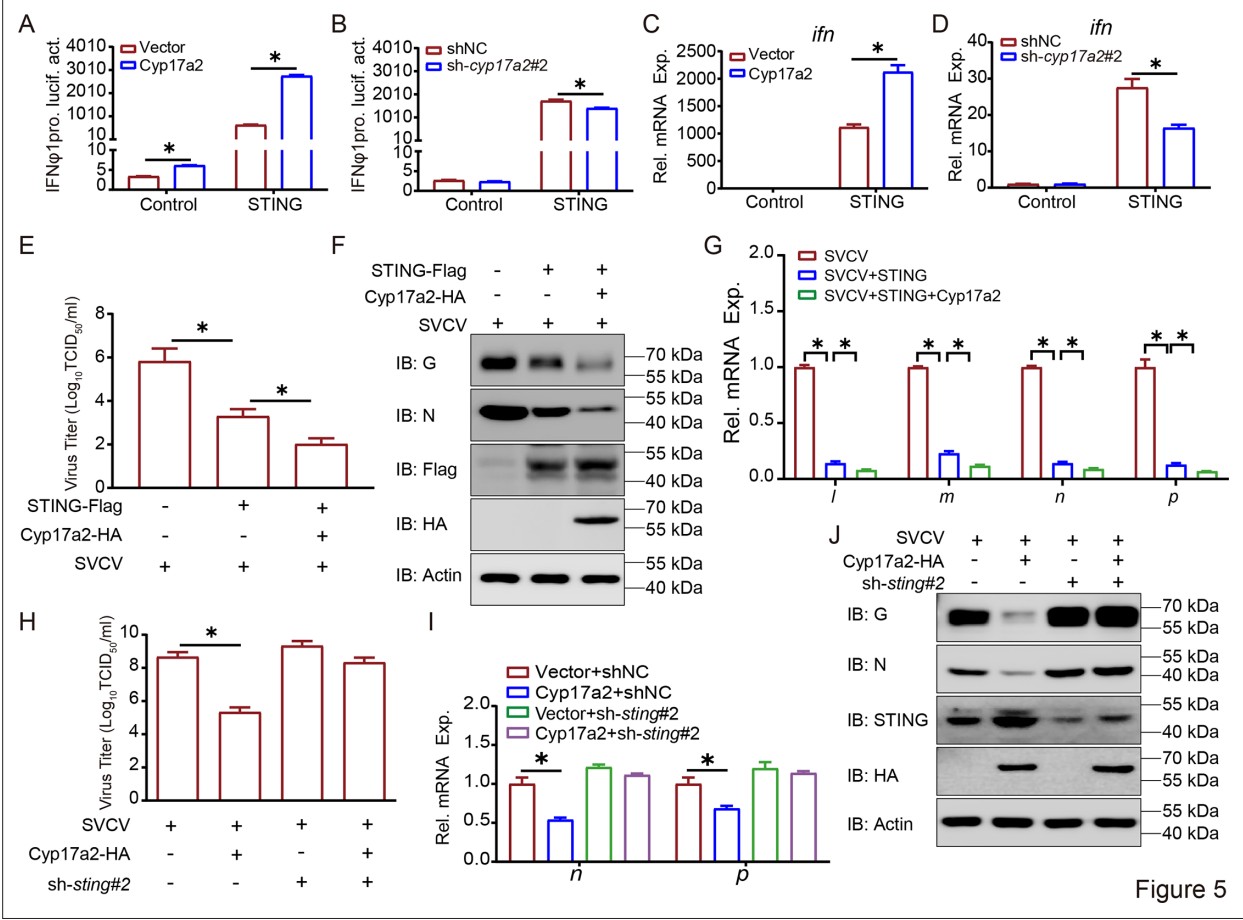

**Figure 5.** Cyp17a2 potentiates STING-induced IFN production and antiviral responses. (**A and B**) Luciferase activity of IFNφ1pro in EPC cells transfected with indicated plasmids for 24 hr before luciferase assays. (**C and D**) qPCR analysis of *ifn* in EPC cells transfected with indicated plasmids for 24 hr. (**E and H**) Detection of virus titers in EPC cells transfected with indicated plasmids for 24 hr and then infected with SVCV (MOI = 1) for 24 hr. (**F and J**) IB analysis of proteins in EPC cells transfected with indicated plasmids for 24 hr and then untreated or infected with SVCV (MOI = 1) for 24 hr. (**G and I**) qPCR analysis of SVCV genes in EPC cells transfected with indicated plasmids for 24 hr, followed by SVCV challenge for 24 hr. All experiments were repeated at least three times with similar results.

The online version of this article includes the following source data and figure supplement(s) for figure 5:

**Source data 1.** PDF file containing original western blots for *Figure 5F and J* indicating the relevant bands and treatments.

**Source data 2.** Original files for western blot analysis displayed in *Figure 5F and J*.

**Source data 3.** Original data for graphs analysis in *Figure 5A–E and G–I*.

**Figure supplement 1.** Cyp17a2 enhances STING-mediated IFN activation and antiviral defense.

**Figure supplement 1—source data 1.** PDF file containing original western blots for *Figure 5—figure supplement 1G* indicating the relevant bands and treatments.

**Figure supplement 1—source data 2.** Original files for western blot analysis displayed in *Figure 5—figure supplement 1G*.

**Figure supplement 1—source data 3.** Original data for graphs analysis in *Figure 5—figure supplement 1A–D, F and I*.

---

revealing btr32 and SOCS3a as STING interactors (*Figure 6D*). btr32 was specifically identified as a Cyp17a2 binding partner through reciprocal Co-IP (*Figure 6E–G*). Then the subcellular localization of btr32 was monitored, and the btr32-GFP signal overlapped with the red signals of the ER marker, indicating that btr32 colocalizes with the ER (*Figure 6H*). Since the results identify that both STING and Cyp17a2 are localized to the ER, btr32 was examined for cellular localization with both STING and Cyp17a2, and the anticipated colocalization pattern was confirmed (*Figure 6I and J*). Cyp17a2 was found to strengthen btr32-STING interaction (*Figure 6K*, *Figure 6—figure supplement 1B*). Functional assays demonstrated that btr32 augmented Cyp17a2-mediated IFNφ1pro activity and *vig1* mRNA induction (*Figure 6L and M*). btr32 potentiated the Cyp17a2-dependent stabilization of

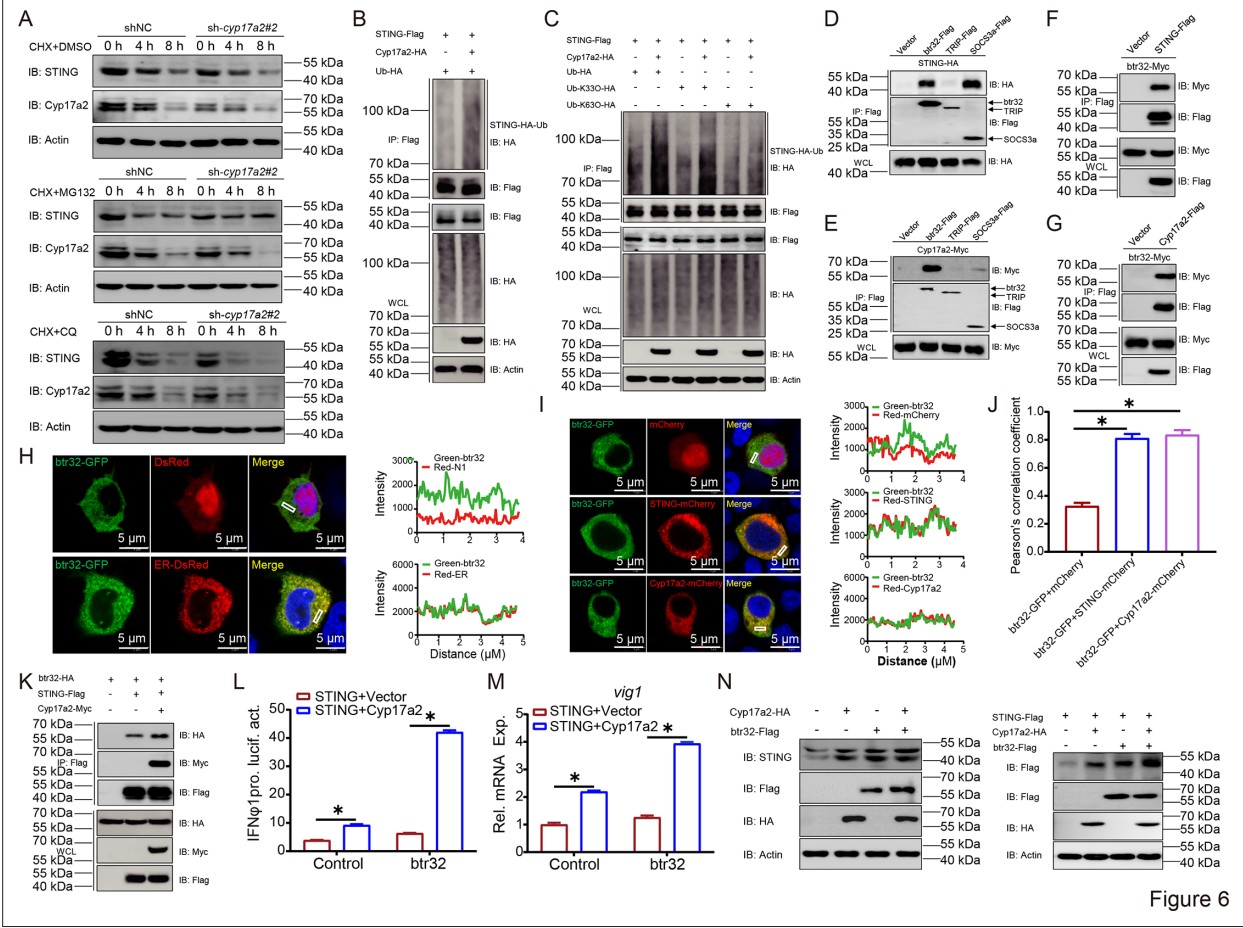

**Figure 6.** Cyp17a2 stabilizes STING by orchestrating btr32-catalyzed K33-linked polyubiquitination. (**A**) IB analysis of proteins in EPC cells transfected with indicated plasmids for 18 hr, then treated with CHX and DMSO, MG132, or CQ for 4 hr and 8 hr. (**B and C**) STING ubiquitination assays in EPC cells transfected with indicated plasmids for 24 hr. (**D–G, K**) IB analysis of WCLs and proteins immunoprecipitated with anti-Flag Ab-conjugated agarose beads from EPC cells transfected with indicated plasmids for 24 hr. (**H and I**) Confocal microscopy of btr32 and ER or STING or Cyp17a2 in EPC cells transfected with indicated plasmids for 24 hr. The coefficient of colocalization was determined by qualitative analysis of the fluorescence intensity of the selected area in Merge. (**A–I, K**) Representative experiments are shown (n=3). (**J**) Colocalization analyses of (**I**) were performed by calculating Pearson correlation coefficient. (**L and M**) Luciferase activity of IFNφ1pro and qPCR analysis of *vig1* in EPC cells transfected with indicated plasmids for 24 hr. (**N**) IB analysis of proteins in EPC cells transfected with indicated plasmids for 24 hr. (**L–N**) Representative experiments are shown (n=3).

The online version of this article includes the following source data and figure supplement(s) for figure 6:

**Source data 1.** PDF file containing original western blots for *Figure 6A–G, K and N* indicating the relevant bands and treatments.

**Source data 2.** Original files for western blot analysis displayed in *Figure 6A–G, K and N*.

**Source data 3.** Original data for graphs analysis in *Figure 6H–J, L and M*.

**Figure supplement 1.** Cyp17a2 promotes STING stabilization by facilitating btr32-catalyzed K33-linked polyubiquitination.

**Figure supplement 1—source data 1.** PDF file containing original western blots for *Figure 6—figure supplement 1A–D and G* indicating the relevant bands and treatments.

**Figure supplement 1—source data 2.** Original files for western blot analysis displayed in *Figure 6—figure supplement 1A–D and G*.

**Figure supplement 1—source data 3.** Original data for graphs analysis in *Figure 6—figure supplement 1E and F*.

STING in both endogenous and exogenous systems (*Figure 6N*). Two shRNAs targeting btr32 were designed and generated, and after validating the knockdown efficiencies for both endogenous and exogenous targets, sh-*btr32*#2 was selected for the subsequent assay (*Figure 6—figure supplement 1C–D*). The knockdown of btr32 impaired Cyp17a2-enhanced STING activity, as shown by reduced IFNφ1pro response and *vig1* transcripts (*Figure 6—figure supplement 1E and F*). Additionally, *btr32* knockdown abolished Cyp17a2-mediated STING upregulation in endogenous and overexpression

systems (*Figure 6—figure supplement 1G*). These findings establish that Cyp17a2 recruits btr32 to catalyze K33-linked polyubiquitination, thereby stabilizing STING protein.

## STING K203 is essential for Cyp17a2-mediated ubiquitination

The mechanism between btr32 and Cyp17a2 in STING stabilization was further characterized. btr32 overexpression was shown to intensify Cyp17a2-driven STING ubiquitination, while *btr32* knockdown attenuated this effect (*Figure 7A*, *Figure 7—figure supplement 1A*). Specific enhancement of K33-linked ubiquitination by btr32 was demonstrated, with reciprocal reduction upon btr32 abrogation (*Figure 7B*, *Figure 7—figure supplement 1B*). Mass spectrometry analysis identified K203 and K221 as candidate ubiquitination sites (*Figure 7C*). Among generated lysine-to-arginine mutants (K203R, K221R), STING stabilization by Cyp17a2 was abolished specifically in the K203R mutant (*Figure 7D*). Corresponding loss of IFN induction and ubiquitination capacity was observed in the K203R mutant (*Figure 7E–G*). K33-linked ubiquitination was specifically impaired in K203R but not K221R variants (*Figure 7—figure supplement 1C*). btr32-mediated enhancement of WT and K33-specific ubiquitination was eliminated in the K203R mutant (*Figure 7H*, *Figure 7—figure supplement 1D*). btr32 domain analysis was performed using RING/BBOX/SPRY deletion mutants (*Figure 7I*). The SPRY domain was required for btr32-STING-Cyp17a2 interactions (*Figure 7J–L*). Both RING and SPRY domains were essential for btr32-driven ubiquitination (*Figure 7M*, *Figure 7—figure supplement 1E*). These data establish K203 as the critical ubiquitination site in STING and define RING/SPRY domains of btr32 as necessary for Cyp17a2-mediated stabilization.

## Cyp17a2 degrades SVCV P protein through modulation of K33-linked polyubiquitination

To determine whether Cyp17a2 directly engages in antiviral activity beyond regulating host immune responses, Co-IP assays were performed. Transient transfection experiments revealed specific interaction between Cyp17a2 and SVCV P protein, which was confirmed reciprocally (*Figure 8A and B*). This interaction was further observed in SVCV-infected cells using P-protein-specific antibody (*Figure 8C*). In our antiviral trials, overexpression of Cyp17a2 led to a reduction in P protein levels during SVCV infection. Consequently, the investigation focused on whether Cyp17a2 could directly target the SVCV P protein for degradation. As anticipated, the co-expression of Cyp17a2 resulted in a decrease in the level of SVCV P protein. Overexpression of Cyp17a2 reduced P protein levels during SVCV infection (*Figure 8D*). Fluorescence microscopy confirmed diminished red fluorescent signals corresponding to SVCV P protein in Cyp17a2-expressing cells (*Figure 8E*). Confocal microscopy analysis of SVCV P protein subcellular distribution revealed that the fluorescent signals from P-mCherry fusion constructs (red) showed partial co-localization with Cyp17a2 (green) in cytoplasmic compartments (*Figure 8F*). To investigate degradation mechanisms, inhibitors targeting distinct pathways were tested. Cyp17a2-mediated P protein degradation was blocked by the proteasome inhibitor MG132, but not by autophagy inhibitors (3-MA, Baf-A1, CQ; *Figure 8G*). Ubiquitination assays demonstrated that Cyp17a2 significantly reduced P protein polyubiquitination (*Figure 8H*). Mutation analysis using ubiquitin variants revealed that Cyp17a2 selectively decreased K33-linked polyubiquitination, as K33R mutants exhibited no significant changes, while K33-specific ubiquitination was markedly reduced (*Figure 8I and J*). In contrast, knockdown of *cyp17a2* exhibited minimal impact on both WT-Ub and K33-linked polyubiquitination levels of the P protein (*Figure 8K*). These findings demonstrate that Cyp17a2 promotes proteasomal degradation of the SVCV P protein by selectively attenuating K33-linked polyubiquitination, defining its role as an antiviral protein.

## The K12 site of P protein is essential for Cyp17a2-mediated degradation

To investigate the specific molecular mechanism underlying ubiquitination-mediated degradation of the P protein, mass spectrometry analysis revealed ubiquitin-specific protease 7 (USP7) and USP8 as candidate deubiquitinating enzymes (*Figure 9A*). Both USP7 and USP8 were found to associate with the P protein; however, only USP8 interacted with Cyp17a2. Therefore, USP8 became the focus of subsequent study (*Figure 9B and C*). The validity of these interactions was further confirmed through reciprocal experiments (*Figure 9—figure supplement 1A*). Confocal microscopy analysis of USP8 subcellular localization demonstrated overlapping cytoplasmic co-localization between USP8-GFP

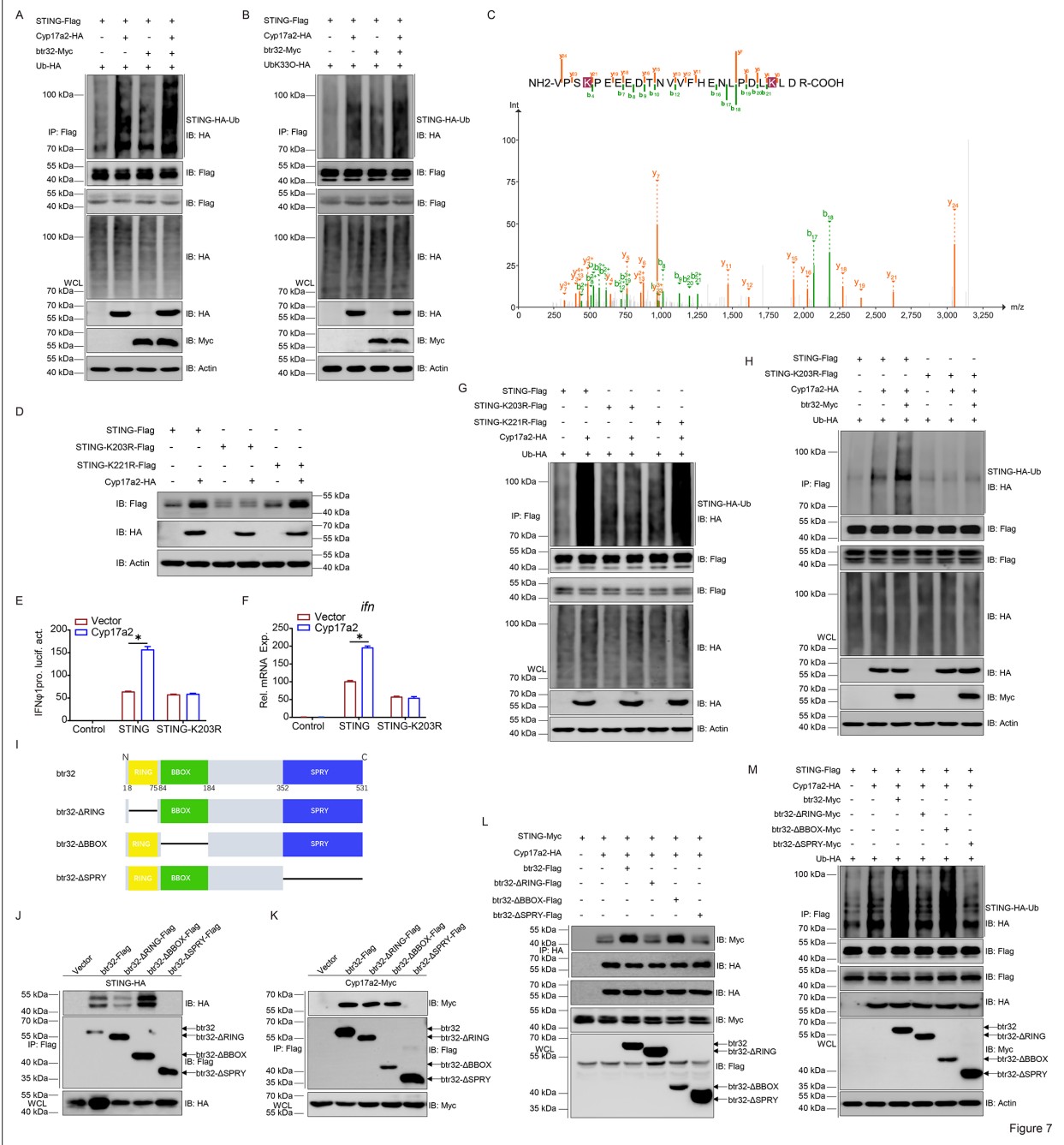

**Figure 7.** Cyp17a2-mediated ubiquitination of STING is dependent on the K203. (**A and B, G and H, and M**) STING ubiquitination assays in EPC cells transfected with indicated plasmids for 24 hr. Representative experiments are shown (n=3). (**C**) Mass spectrometry analysis of a peptide derived from ubiquitinated STING-Myc. (**D**) IB analysis of proteins in EPC cells transfected with indicated plasmids for 24 hr. (**E and F**) Luciferase activity of IFNφ1pro and qPCR analysis of *ifn* in EPC cells transfected with indicated plasmids for 24 hr. (**D–F**) Representative experiments are shown (n=3). (**I**) Schematic representation of full-length btr32 and its mutants. (**J–L**) IB of WCLs and proteins immunoprecipitated with anti-Flag/HA Ab-conjugated agarose beads from EPC cells transfected with indicated plasmids for 24 h. Representative experiments are shown (n=3).

The online version of this article includes the following source data and figure supplement(s) for figure 7:

**Source data 1.** PDF file containing original western blots for *Figure 7A, B, D, G, H and J–M* indicating the relevant bands and treatments.

**Source data 2.** Original files for western blot analysis displayed in *Figure 7A, B, D, G, H and J–M*.

**Source data 3.** Original data for graphs analysis in *Figure 7E and F*.

**Figure supplement 1.** The K203 residue is essential for the ubiquitination of STING mediated by Cyp17a2.

*Figure 7 continued on next page*

Figure 7 continued

**Figure supplement 1—source data 1.** PDF file containing original western blots for *Figure 7—figure supplement 1A–E* indicating the relevant bands and treatments.

**Figure supplement 1—source data 2.** Original files for western blot analysis displayed in *Figure 7—figure supplement 1A–E*.

(green fluorescence) and mCherry-tagged Cyp17a2 or the P protein (red fluorescence; *Figure 9D*, *Figure 9—figure supplement 1B*). Cyp17a2 overexpression enhanced USP8 and P protein binding, whereas *cyp17a2* knockdown disrupted this interaction (*Figure 9E*, *Figure 9—figure supplement 1C*). USP8 overexpression amplified Cyp17a2-mediated P protein reduction (*Figure 9F*). Knockdown

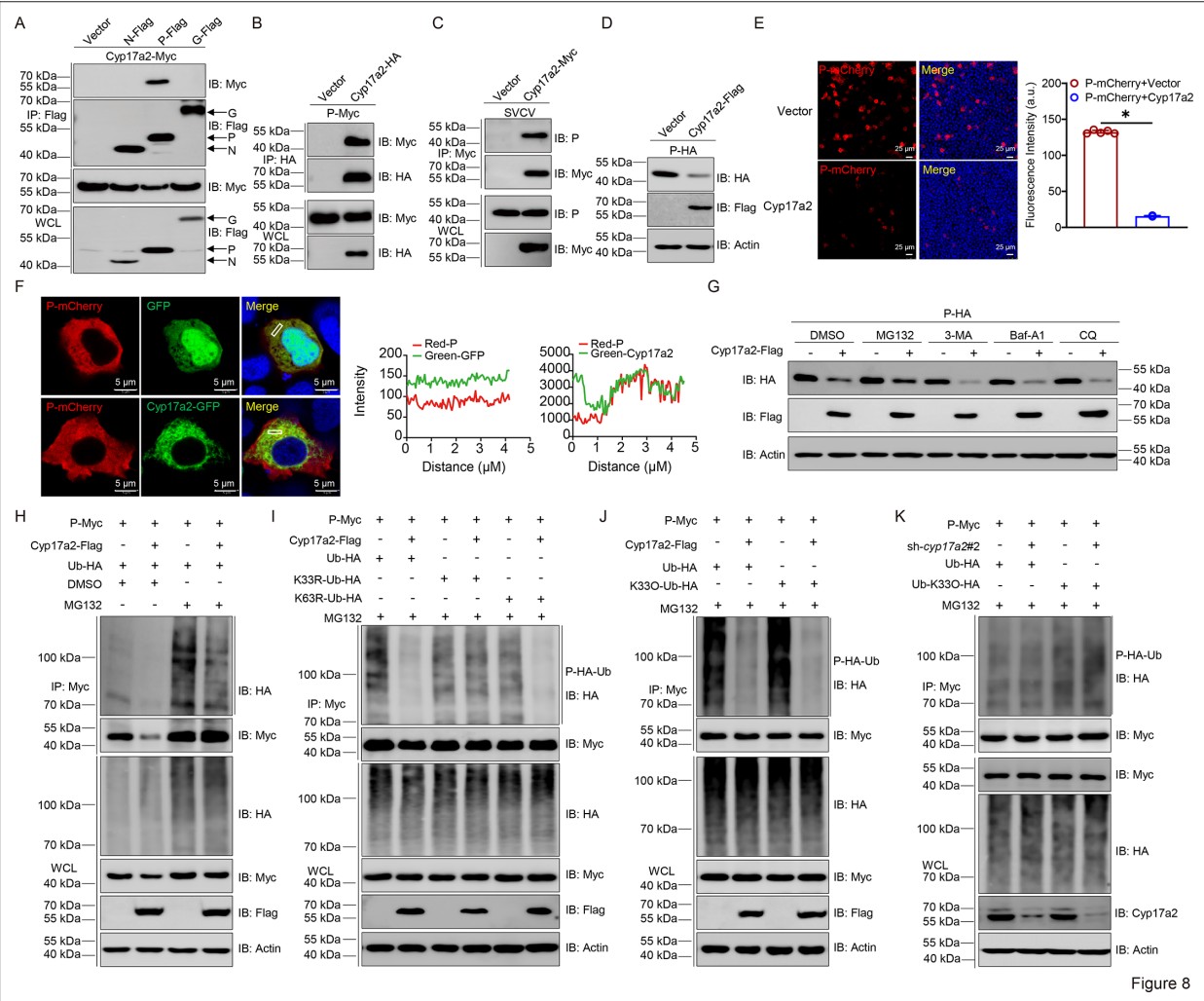

**Figure 8.** Cyp17a2 induces proteasomal degradation of the SVCV P protein by catalyzing K33-linked polyubiquitination. (**A–C**) IB analysis of WCLs and proteins immunoprecipitated with anti-Flag, anti-HA, or anti-Myc Ab-conjugated agarose beads from EPC cells transfected with indicated plasmids for 24 hr. (**D**) IB analysis of proteins in EPC cells transfected with indicated plasmids for 24 hr. (**E**) Fluorescent analysis of proteins in EPC cells transfected with indicated plasmids for 24 hr. (**F**) Confocal microscopy of P and Cyp17a2 in EPC cells transfected with indicated plasmids for 24 hr. The coefficient of colocalization was determined by qualitative analysis of the fluorescence intensity of the selected area in Merge. (**G**) IB analysis of proteins in EPC cells transfected with indicated plasmids for 18 hr, followed by treatments of MG132 (10 μM), 3-MA (2 mM), Baf-A1 (100 nM), and CQ (50 μM) for 6 hr, respectively. (**H–K**) P protein ubiquitination assays in EPC cells transfected with indicated plasmids for 18 hr, followed by MG132 treatments for 6 hr. All experiments were repeated at least three times with similar results.

The online version of this article includes the following source data for figure 8:

**Source data 1.** PDF file containing original western blots for *Figure 8A–D and G–K* indicating the relevant bands and treatments.

**Source data 2.** Original files for western blot analysis displayed in *Figure 8A–D and G–K*.

**Source data 3.** Original data for graphs analysis in *Figure 8E and F*.

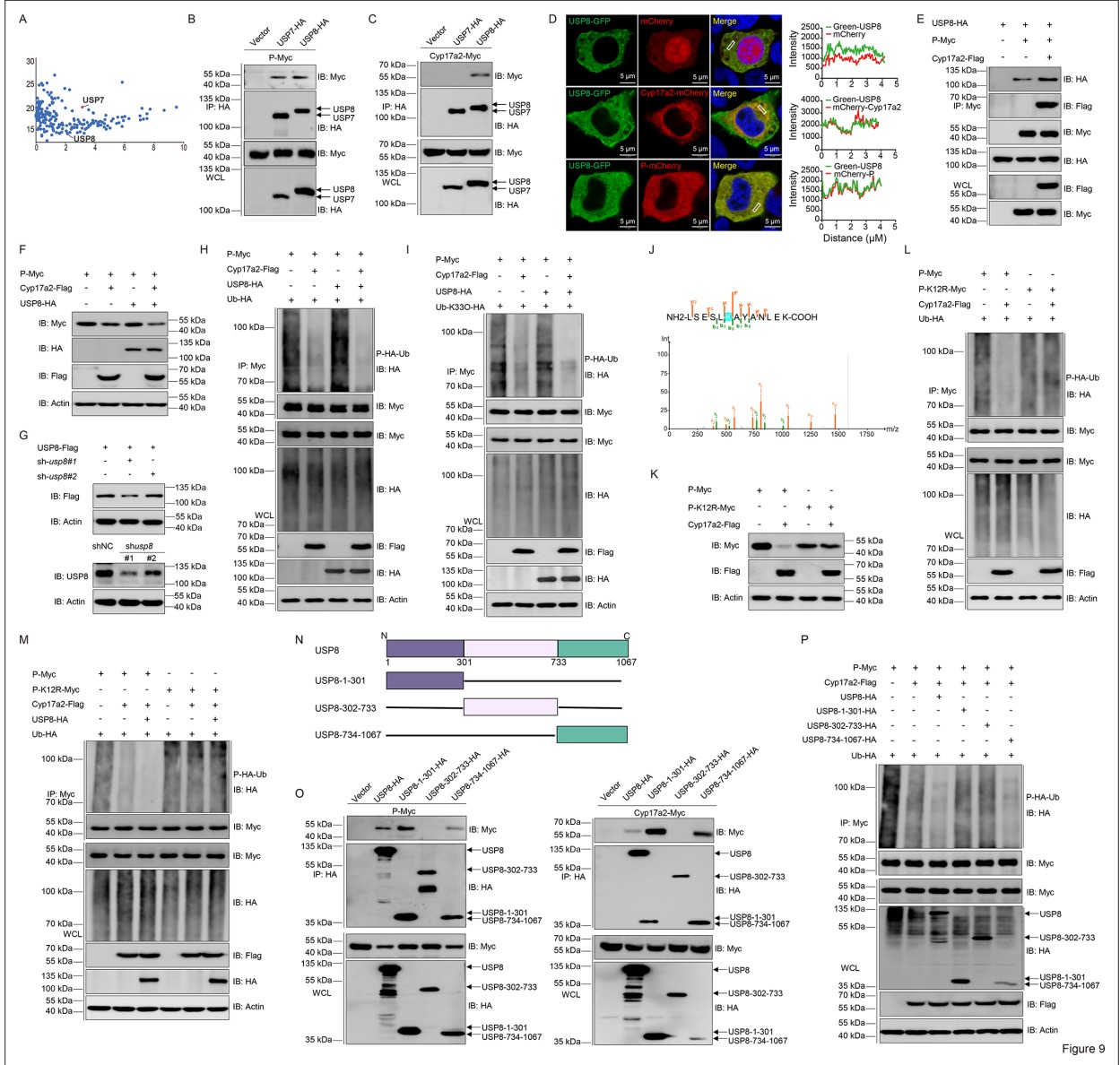

**Figure 9.** Cyp17a2 targets the K12 site of the P protein for K33-linked polyubiquitination. (**A**) List of proteins interacting with P protein detected by mass spectrometry. (**B and C, E, O**) IB analysis of WCLs and proteins immunoprecipitated with anti-HA/Myc Ab-conjugated agarose beads from EPC cells transfected with indicated plasmids for 24 hr. Representative experiments are shown (n=3). (**D**) Confocal microscopy of USP8 and Cyp17a2 or P protein in EPC cells transfected with indicated plasmids for 24 hr. The coefficient of colocalization was determined by qualitative analysis of the fluorescence intensity of the selected area in Merge. (**F and G, K**) IB analysis of proteins in EPC cells transfected with indicated plasmids for 24 hr. (**H and I, L and M, P**) P protein ubiquitination assays in EPC cells transfected with indicated plasmids for 18 hr, followed by MG132 treatments for 6 hr. (**F-I, K-M, and P**) Representative experiments are shown (n=3). (**J**) Mass spectrometry analysis of a peptide derived from ubiquitinated P-Myc. (**N**) Schematic representation of full-length USP8 and its mutants.

The online version of this article includes the following source data and figure supplement(s) for figure 9:

**Source data 1.** PDF file containing original western blots for *Figure 9B, C, E–I, K–M, O and P* indicating the relevant bands and treatments.

**Source data 2.** Original files for western blot analysis displayed in *Figure 9B, C, E–I, K–M, O and P*.

**Source data 3.** Original data for graphs analysis in *Figure 9D*.

**Figure supplement 1.** Cyp17a2 mediates K33-linked polyubiquitination of the P protein at K12.

**Figure supplement 1—source data 1.** PDF file containing original western blots for *Figure 9—figure supplement 1A and C–I* indicating the relevant bands and treatments.

**Figure supplement 1—source data 2.** Original files for western blot analysis displayed in *Figure 9—figure supplement 1A and C–I*.

**Figure supplement 1—source data 3.** Original data for graphs analysis in *Figure 9—figure supplement 1B*.

of USP8 abolished Cyp17a2-dependent P protein degradation (*Figure 9G*, *Figure 9—figure supplement 1D*). USP8 facilitated Cyp17a2-induced deubiquitination of P protein, with knockdown reversing this effect (*Figure 9H*, *Figure 9—figure supplement 1E*). This result was also observed in the K33-linked ubiquitination assay (*Figure 9I*, *Figure 9—figure supplement 1F*). Mass spectrometry identified K12 as a critical lysine modification site (*Figure 9J*). The K12R mutant resisted Cyp17a2-mediated degradation and showed impaired deubiquitination in WT ubiquitin and K33-linked assays (*Figure 9K-L*, *Figure 9—figure supplement 1G*). USP8 failed to enhance Cyp17a2-mediated deubiquitination in K12R mutants (*Figure 9M*, *Figure 9—figure supplement 1H*). Truncation analysis demonstrated that USP8 residues 1–301 aa or 734–1067 aa were essential for interactions with P protein and Cyp17a2 (*Figure 9N and O*). USP8 mutants only residues 1–301 aa or 302–733 aa lost the ability to support Cyp17a2-mediated P protein deubiquitination (*Figure 9P*, *Figure 9—figure supplement 1I*). These data establish that Cyp17a2 requires USP8 and the P protein K12 site to mediate deubiquitination-dependent degradation, with USP8 residues 734–1067 aa being indispensable for this process.

## Discussion

The influence of biological sex on antiviral immunity shows both evolutionary parallels and divergences across species. While mammals typically exhibit stronger female immunity through sex chromosome and hormonal mechanisms, our study reveals an intriguing reversal in zebrafish, where males show enhanced antiviral responses via an autosomal gene-mediated pathway. This supplements conventional views and highlights the evolutionary plasticity of teleost immune systems.

Teleosts display remarkable diversity in sex determination mechanisms, including XX/XY, ZZ/ZW, and environmental systems (*Matsuda et al., 2002*; *Yoshida et al., 2011*; *Hayashi et al., 2010*). This diversity provides a unique framework for studying sexual dimorphism independent of sex chromosomes. Using zebrafish, which lack conserved sex chromosomes, allows us to exclude the confounding effects of divergent sex-chromosome gene dosage and focus on the role of autosomal genes and their differential regulation in shaping sex-biased immunity. Our study leverages this unique context to demonstrate that enhanced antiviral immunity in males is mediated by the male-biased expression of the autosomal gene *cyp17a2*.

Mechanistically, Cyp17a2 operates through dual antiviral pathways; it stabilizes STING via btr32-mediated K33-linked ubiquitination, enhancing IFN production, and recruits USP8 to remove K33-linked chains from viral P protein, promoting its degradation (*Figure 10*). This establishes Cyp17a2 as a central regulator of sex-specific antiviral immunity independent of sex chromosome

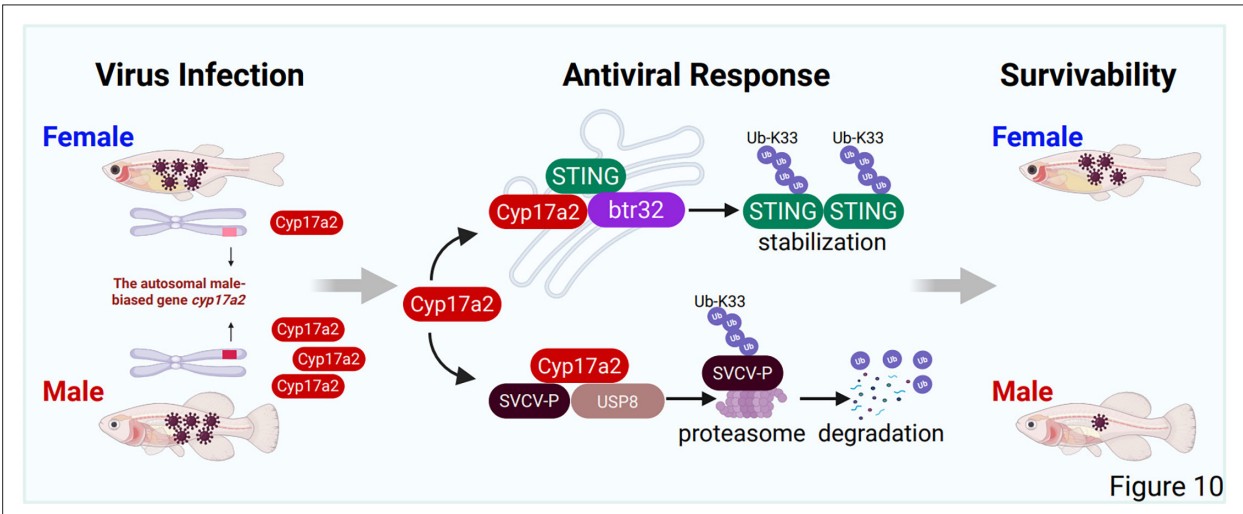

**Figure 10.** A mechanistic model illustrating the dual regulatory roles of Cyp17a2 in upregulating STING expression and degrading SVCV P protein. Upon virus infection, male-biased autosomal gene *cyp17a2* enhances antiviral responses in males. Mechanistically, Cyp17a2, an ER-localized protein, stabilizes STING expression through recruitment of the E3 ubiquitin ligase btr32, promoting K33-linked polyubiquitination. Furthermore, Cyp17a2 facilitates proteasomal degradation of the SVCV P protein by engaging USP8 to reduce its K33-linked polyubiquitination, thereby amplifying antiviral IFN responses.

pathways. Particularly noteworthy is the demonstration that sexual dimorphism in immunity can arise through differential regulation of autosomal genes rather than direct sex chromosome effects - a mechanism distinct from mammalian X-linked immune gene dosage effects or estrogen-mediated immunomodulation.

A key question emerging from our findings is whether the Cyp17a2-mediated antiviral mechanism is conserved across the teleost lineage or represents an adaptation specific to cyprinids such as zebrafish. The teleost-specific nature of *cyp17a2* suggests functions shaped by aquatic environmental challenges. While further studies in diverse teleost families are needed, the presence of *cyp17a2* orthologs across fish genomes indicates potential conservation. It is proposed that the recruitment of a core component of the steroidogenic enzyme machinery into the innate immune network may represent a significant evolutionary adaptation in teleosts. Aquatic environments have been shown to have a persistently high pathogen load. The co-option of a gene such as *cyp17a2*, which is already subject to sex-biased regulatory control, offers an efficient and rapidly deployable strategy for establishing sexually dimorphic immunity, obviating the necessity for de novo evolution of entirely new pathways.

Sexual dimorphism in viral susceptibility represents a conserved yet mechanistically complex phenomenon across vertebrates (*Zajitschek et al., 2020*). While males generally exhibit increased vulnerability to numerous viral pathogens such as dengue virus and influenza A virus, notable exceptions exist where females demonstrate higher susceptibility, including infections by herpes simplex virus and measles virus (*Klein, 2012*). These divergent patterns underscore a dual dependency of sex-biased infection outcomes on both host immunological determinants and viral evolutionary strategies (*Klein and Huber, 2010*). For instance, the male predominance observed in severe dengue virus cases may be explained by both gender-specific occupational exposure risks and sexual dimorphism in Ab-dependent enhancement mechanisms, potentially linked to androgen-mediated modulation of antiviral immune responses (*Gan et al., 2017*). Whereas female susceptibility to herpes simplex virus could be related to ovarian sex hormones, including estradiol and progesterone (*Wald, 2004*). Critically, sex differences in antiviral immunity are not static but evolve across the lifespan and are tightly linked to age and reproductive status (*Klein and Flanagan, 2016*). For example, sex disparities in hantavirus infection outcomes emerge post-puberty, coinciding with hormonal maturation (*Vial et al., 2023*). Similarly, during the 2009 H1N1 pandemic in Japan, morbidity rates exhibited a triphasic sex bias: males <20 and >80 years old showed higher susceptibility than females, whereas females of reproductive age (20–49 years) faced elevated risks (*Eshima et al., 2011*). This pattern aligns with the hypothesis that estrogen peaks during reproductive years enhance antiviral defenses in females, while age-related declines in gonadal hormones may diminish this protective effect in postmenopausal populations (*Cui et al., 2013*). Our study extends these principles to teleost species, revealing novel sex-specific resistance patterns in fish-virus interactions. Male zebrafish demonstrated heightened resistance to SVCV, while male crucian carp exhibited reduced susceptibility to CyHV-2 (*Lu et al., 2022*). These findings suggest that male-biased resistance may represent an evolutionarily conserved strategy in certain fish-virus systems. This may reflect evolutionary adaptations to unique aquatic pressures. We propose several non-mutually exclusive hypotheses, which are rooted in the unique life histories and ecological pressures faced by aquatic vertebrates (*Gui et al., 2022*). Firstly, the aquatic environment acts as a potent reservoir for pathogens, constantly exposing fish to diverse microbial challenges. This high pathogen pressure may select for specialized and potent immune strategies. If males and females encounter different levels or types of parasites or infections due to their behavior or morphology, this could lead to the evolution of sex-specific immune adaptations. Secondly, this phenomenon can be viewed through the lens of life-history trade-offs. Investment in immunity is costly in terms of energy and must be balanced against other demands, such as reproduction. In many teleost species, male reproductive success depends heavily on courtship displays, nest guarding, and aggressive competition with other males, all of which are high-energy activities that can compromise immune function. Finally, the prevalence of external fertilization in teleosts is a crucial factor. Unlike mammals, where the embryo develops internally and is protected by the maternal immune system, fish embryos are exposed to the pathogen-rich aquatic environment from the moment of fertilization. Therefore, a male's contribution to offspring survival includes not only his genetic material, but also the intrinsic pathogen resistance of his sperm and the antimicrobial properties of the seminal fluid or nest site.

STING was originally identified as a critical adaptor molecule mediating antiviral signaling cascades (*Woznica et al., 2021*; *Ran et al., 2014*). Subsequential studies have established its indispensable role in orchestrating innate immune responses against both DNA and RNA viral pathogens. Current mechanistic understanding reveals that upon detection of cytoplasmic pathogen-derived DNA, STING operates through two distinct molecular pathways: serving as a downstream effector of essential single-stranded/double-stranded DNA sensors or functioning as a direct receptor for microbial cyclic dinucleotides (CDNs). Following activation, this TM protein recruits and activates TBK1, which subsequently phosphorylates IRF3 to facilitate its nuclear translocation and initiate IFN production. Notably, the protein demonstrates functional pleiotropy in RNA viral infections through selective interaction with RIG-I rather than melanoma differentiation-associated protein 5 (MDA5), thereby activating parallel IRF3-dependent antiviral signaling pathways. Numerous studies indicate remarkable conservation of STING-mediated IFN induction mechanisms across teleost species (*Biacchesi et al., 2012*; *Lu et al., 2023*; *Li et al., 2024*). Mechanism studies further reveal that fish STING maintains dual antiviral functionality against both nucleic acid virus types, while simultaneously being subjected to targeted manipulation by various viral pathogens as an immune evasion strategy.

Beyond its steroidogenic functions, Cyp17a2 represents a novel scaffold protein in ubiquitination pathways. Despite the absence of clearly identifiable domains that are directly implicated in ubiquitination (e.g. RING, HECT, or U-box domains), it is proposed that the subcellular localization and protein-interaction capability of Cyp17a2 are pivotal to its novel function. As an ER-resident protein, Cyp17a2 is strategically co-localized with central innate immune sensors such as STING. This shared localization facilitates specific protein-protein interactions, enabling Cyp17a2 to serve as an essential platform that recruits the E3 ubiquitin ligase btr32 to stabilize STING via K33-linked ubiquitination. In contrast, Cyp17a2 engages the deubiquitinate USP8 to remove K33-linked chains from the viral P protein, thereby targeting it for degradation. This bifunctional and scaffolding role, which orchestrates both the addition and removal of specific ubiquitin chains to fine-tune antiviral responses, represents a fascinating evolutionary adaptation in teleost.

The CYP450 enzyme superfamily exhibits multifaceted roles across species, encompassing physiological processes and immune regulation. In human immune cells, pharmacological inhibition of CYP1A has been demonstrated to upregulate the expression of stem cell factor receptor and interleukin 22, directly linking CYP1-dependent metabolism of environmental small molecules to immune modulation (*Effner et al., 2017*). Evolutionary conservation of this regulatory mechanism is evident in teleost, where grass carp (*Ctenopharyngodon idella*) CYP1A demonstrates dynamic expression during grass carp reovirus (GCRV) infection, implicating its role in antiviral immunity (*Chu et al., 2019*). Our study further identifies Cyp17a2 as a sexually dimorphic regulator of IFN signaling in fish. This finding expands the recognized roles of the CYP450 superfamily in immune regulation, revealing a novel mechanism underlying sex-specific immunity in lower vertebrates.

In conclusion, the zebrafish model with its lack of conserved sex chromosomes and environmental sex determination plasticity provides unique insights into the evolutionary continuum of sexual dimorphism. Our results suggest that the relationship between sexual differentiation and immune competence may be more fluid in teleosts than in mammals, potentially reflecting adaptive responses to aquatic pathogens and variable mating systems. Future investigations should explore whether this autosomal-centered mechanism represents a teleost-specific adaptation or an ancestral feature subsequently modified in terrestrial vertebrates, while also examining potential interactions between environmental sex determinants and immune gene regulation.

## Materials and methods
### Fish, cells, and viruses

Mature zebrafish individuals 2.5 months after hatching (0.4±0.1 g) were selected in this study. Wild-type (WT) zebrafish (AB strain; *Danio rerio*) were procured from the China Zebrafish Resource Center (CZRC), while *cyp17a2* mutant lines were generously provided by Dr. Zhan Yin's laboratory at the Institute of Hydrobiology, Chinese Academy of Sciences. All specimens were bred using standardized procedures. In accordance with the ethical requirements and national animal welfare guidelines, all experimental fish were required to undergo a two-week acclimatization period in the laboratory and have their health assessed prior to the commencement of the study. Only fish that appeared

healthy and were mobile were selected for scientific investigation. Zebrafish embryonic fibroblast cells (ZF4) (RRID:CVCL_3275) (American Type Culture Collection, ATCC) were cultured in Ham's F-12 medium (Thermo Scientific, 11765054) supplemented with 10% fetal bovine serum (FBS) (Vivacell, C04001-500) at 28 °C and 5% $CO_2$. *Epithelioma papulosum cyprini* (EPC) cells (RRID:CVCL_6E02) were obtained from the Chinese Culture Collection Centre for Type Cultures (CCTCC), Gibel carp brain (GiCB) cells (RRID:CVCL_CW64) were provided by Ling-Bing Zeng (Yangtze River Fisheries Research Institute, Chinese Academy of Fishery Sciences), these cells were maintained at 28 °C in 5% $CO_2$ in medium 199 (Thermo Scientific, 11150067) supplemented with 10% FBS. All cell lines were routinely tested for mycoplasma. Spring viremia of carp virus (SVCV) was propagated in EPC cells until a CPE was observed, and then cell culture fluid containing SVCV was harvested and centrifuged at $4\times10^3$ g for 20 min to remove the cell debris, and the supernatant was stored at –80 °C until used. Cyprinid herpesvirus 2 (CyHV-2, obtained from Yancheng city, Jiangsu province, China) was provided by Prof. Liqun Lu (Shanghai Ocean University). CyHV-2 was propagated in GICB cells and harvested in a similar way to SVCV.

## Plasmid construction and reagents

The sequences of zebrafish and gibel carp (*Carassius gibelio*) Cyp17a2 (GenBank accession number: NM_001105670.1 and XM_026199804.1) were obtained from the National Centre for Biotechnology Information (NCBI) website. Cyp17a2 was amplified by polymerase chain reaction (PCR) using cDNA from adult zebrafish or gibel carp tissues as a template and cloned into the expression vector pCMV-HA (Cat#631604) or pCMV-Myc (Cat#K6003-1) (Clontech) vectors. Zebrafish MAVS (NM_001080584.2), TBK1 (NM_001044748.2), STING (NM_001278837.1) and the truncated mutants of STING, IRF3 (NM_001143904.1), IRF7 (BC058298.1), btr32 (XM_021470074.1) and the truncated mutants of btr32, TRAIP (NM_205607.1), SOCS3a (NM_199950.1), USP7 (XM_021473871.1), USP8 (XM_009293353.4), and N, P, and G protein (DQ097384.2) of SVCV were cloned into pCMV-Myc and pCMV-Tag2C vectors. The shRNA of *Pimephales promelas* Cyp17a2 (XM_039679078.1), STING (HE856620.1), and USP8 (XM_039680290.1) was designed by BLOCK-iT RNAi Designer and cloned into the pLKO.1-TRC Cloning vector (RRID:Addgene_10878). For subcellular localization experiments, Cyp17a2 was constructed onto pEGFP-N3 (Clontech), while STING and P protein were constructed onto pCS2-mCherry (Clontech). The plasmids containing zebrafish IFNφ1pro-Luc and ISRE-Luc in the pGL3-Basic luciferase reporter vector (Promega) were constructed as described previously. The *Renilla* luciferase internal control vector (pRL-TK) was purchased from Promega. The ubiquitin mutant expression plasmids Lys-33/63 (all lysine residues were mutated except Lys-33 or Lys-63) and Lys-33R/63 R (only Lys-33 or Lys-63 mutated) were ligated into the pCMV-HA vectors named Ub-K33O/K63O-HA and Ub-K33R/K63R. All constructs were confirmed by DNA sequencing. Polyinosinic-polycytidylic acid (poly I:C) was purchased from Sigma-Aldrich (P0913) used at a final concentration of 1 µg/µl. MG132 (M7449), 3-Methyladenine (3-MA) (M9281), CQ (C6628) was obtained from Sigma-Aldrich. Bafilomycin A1 (Baf-A1) (S1413) and CHX (NSC-185) were obtained from Selleck.

## Transcriptomic analysis

Total RNA was extracted using the TRIzol method and evaluated for RNA purity and quantification using a NanoDrop 2000 spectrophotometer (Thermo Fisher Scientific, Waltham, U.S.A.) and RNA integrity was assessed using an Agilent 2100 Bioanalyzer (Agilent Technologies, Santa Clara, U.S.A.). Transcriptome sequencing and data analysis were conducted by OE Biotech (Shanghai, China). The raw sequencing data was submitted to the NCBI (GEO accession number: GSE286486).

## Transient transfection and virus infection

EPC cells were transfected in 6-well and 24-well plates using transfection reagents from FishTrans (MeiSenTe Biotechnology) according to the manufacturer's protocol. Antiviral assays were conducted in 24-well plates by transfecting EPC cells with the plasmids shown in the figure. At 24 hr post-transfection, cells were infected with SVCV at a multiplicity of infection (MOI = 0.001). After 24 hr, supernatant aliquots were harvested for detection of virus titers, the cell monolayers were fixed by 4% paraformaldehyde (PFA) and stained with 1% crystal violet for visualizing CPE. For virus titration, 200 µl of culture medium were collected at 48 hr post-infection and used for detection of virus titers according to the method of Reed and Muench. The supernatants were subjected to threefold or

10-fold serial dilutions and then added (100 µl) onto a monolayer of EPC cells cultured in a 96-well plate. After 48 or 72 hr, the medium was removed and the cells were washed with phosphate-buffered saline (PBS), fixed by 4% PFA, and stained with 1% crystal violet. The virus titer was expressed as 50% tissue culture infective dose ($TCID_{50}$/ml). For viral infection, fish were anesthetized with meth-anesulfonate (MS-222) and intraperitoneally (i.p.) injected with 5 µl of M199 containing SVCV ($5 \times 10^8$ $TCID_{50}$/ml). The i.p. injection of PBS was used as mock infection. Then the fish were migrated into the aquarium containing new aquatic water.

## Luciferase activity assay

EPC cells were cultured overnight in 24-well plates and subsequently co-transfected with the expression plasmid and luciferase reporter plasmid. The cells were infected with SVCV or transfected with poly I:C for 24 hr prior to harvest. At 24 hr post-stimulation, cells were washed with PBS and lysed for measuring luciferase activity by the Dual-Luciferase Reporter Assay System (Promega) according to the manufacturer's instructions. Firefly luciferase activity was normalized based on the *Renilla* luciferase activity.

## RNA extraction, reverse transcription, and quantitative PCR (qPCR)

The RNA was extracted using TRIzol Reagent (Thermo Fisher Scientific, 10177091), and first-strand cDNA was synthesized with a PrimeScript RT kit with gDNA Eraser (Takara). qPCR was performed on the CFX96 Real-Time System (Bio-Rad) using SYBR green PCR Master Mix (Yeasen). The PCR conditions were as follows: 95 °C for 5 min and then 40 cycles of 95 °C for 20 s, 60 °C for 20 s, and 72 °C for 20 s. The primers utilized for the qPCRs are presented in *Supplementary file 1*, and the *β-actin* gene was utilized as the internal control. The relative fold changes were calculated by comparison to the corresponding controls using the $2^{-\Delta\Delta Ct}$ method (where CT was the threshold cycle).

## Co-immunoprecipitation (Co-IP) assay

EPC cells were cultured in 10 cm dishes overnight and subsequently transfected with 10 µg plasmid as illustrated. At 24 hr post-transfection, the medium was removed and cells were washed with PBS. Then the cells were lysed in 1 ml radioimmunoprecipitation (RIPA) lysis buffer [1% NP-40, 50 mM Tris-HCl, pH 7.5, 150 mM NaCl, 1 mM EDTA, 1 mM NaF, 1 mM sodium orthovanadate ($Na_3VO_4$), 1 mM phenyl-methylsulfonyl fluoride (PMSF), 0.25% sodium deoxycholate] containing protease inhibitor cocktail (Sigma-Aldrich) at 4 °C for 1 hr on a rocker platform. The cellular debris was removed by centrifugation at 12,000 × g for 15 min at 4 °C. The supernatant was transferred to a fresh tube and incubated with 20 µl anti-Flag/HA/Myc affinity gel (Sigma-Aldrich, A2220/E6779/E6654) overnight at 4 °C with constant rotating incubation. These samples were further analyzed by immunoblotting (IB). Immunoprecipitated proteins were collected by centrifugation at 5000 × g for 1 min at 4 °C, washed three times with lysis buffer, and resuspended in 50 µl 2×SDS sample buffer. The immunoprecipitates and whole cell lysates (WCLs) were analyzed by IB with the indicated antibodies (Abs).

## In vivo ubiquitination assay

The transfected EPC cells were washed twice with 10 ml ice-cold PBS and subsequently digested with 1 ml 0.25% trypsin-EDTA (1×) (Thermo Scientific, 25200072) for 2–3 min until the cells were dislodged. 100 µl FBS was added to neutralize the trypsin and the cells were resuspended into 1.5 ml centrifuge tube, centrifuged at 2000 × g for 5 min. The supernatant was discarded, and the cell precipitations were resuspended using 1 ml PBS and centrifuged at 2000 × g for 5 min. The collected cell precipitations were lysed using 100 µl PBS containing 1% SDS and denatured by heating for 10 min. The supernatants were diluted with lysis buffer until the concentration of SDS was reduced to 0.1%. The diluted supernatants were incubated with 20 µl anti-Myc affinity gel overnight at 4 °C with constant agitation. Subsequently, these samples were subjected to further analysis by IB. The immunoprecipitated proteins were collected by centrifugation at 5000 × g for 1 min at 4 °C, washed three times with lysis buffer, and resuspended in 100 µl 1×SDS sample buffer.

## Immunoblot analysis

Immunoprecipitates or WCLs were analyzed as described previously. Abs were diluted as follows: anti-*β-actin* (ABclonal, AC026) at 1:10000, anti-Flag (Sigma-Aldrich, F1804) at 1:3000, anti-HA (Covance,

MMS-101R) at 1:3000, anti-Myc (Santa Cruz Biotechnology, sc-40) at 1:3000, anti-btr32 (ABclonal, A13887) at 1:1000, anti-USP8 (proteintech, 27791–1-AP) at 1:1000, and HRP-conjugated anti-mouse/rabbit IgG (Thermo Scientific, 31430/31460) at 1:5000, anti-N/P/G/STING/Cyp17a2 (prepared and purified in our lab) at 1:2000.

## Immunofluorescence (IF)

EPC cells were plated onto glass coverslips in six-well plates and infected with SVCV (MOI = 1) 24 hr. The cells were then washed with PBS and fixed in 4% PFA at room temperature for 1 hr and permeabilized with 0.2% Triton X-100 in ice-cold PBS for 15 min. The samples were incubated for 1 hr at room temperature in PBS containing 2% bovine serum albumin (BSA, Sigma-Aldrich, V900933). After additional PBS washing, the samples were incubated with anti-N Ab in PBS containing 2% BSA for 2–4 hr at room temperature. After being washed three times by PBS, the samples were incubated with secondary Ab (Goat anti-Rabbit IgG (H+L) Highly Cross-Adsorbed Secondary Antibody, Alexa Fluor Plus 488) (Thermo Scientific, A32731, 1:5000) in PBS containing 2% BSA for 1 hr at room temperature. After additional PBS washing, the cells were finally stained with 1 µg/ml 4′, 6-diamidino-2-phenylindole (DAPI, Beyotime Institute of Biotechnology, C1006) for 10 min in the dark at room temperature. Finally, the coverslips were washed and observed with a confocal microscope under a 10×immersion objective (SP8; Leica).

## Fluorescent microscopy

EPC cells were plated onto coverslips in six-well plates and transfected with the plasmids indicated for 24 hr. Thereafter, the cells were washed twice with PBS and fixed with 4% PFA for 1 hr. After being washed three times with PBS, the cells were stained with 1 µg/ml DAPI for 15 min in the dark at room temperature. Subsequently, the coverslips were washed and observed with a confocal microscope under a 63×oil immersion objective (SP8; Leica).

## Histopathology

Liver and spleen tissues from three individuals of control or virus-infected fish at 2 days post infection (dpi) were dissected and fixed in Bouin's Fixative overnight. Subsequently, the samples were dehydrated in ascending grades of alcohol and embedded into paraffin. Sections at 5 µm thickness were taken and stained with hematoxylin and eosin (H&E). Histological changes were examined by optical microscopy at ×40 magnification and were analyzed by the Aperio ImageScope software.

## Statistics analysis

For fish survival analysis, Kaplan-Meier survival curves were generated and subsequently subjected to a log-rank test. For the bar graph, one representative experiment of at least three independent experiments is shown, and each was done in triplicate. For the dot plot graph, each dot point represents one independent biological replicate. Unpaired Student's t test was used for statistical analysis. Data are presented as mean ± standard error of the mean (SEM). A p-*value* p-value <0.05 was considered statistically significant.

# Acknowledgements

We thank Fang Zhou (Institute of Hydrobiology, Chinese Academy of Sciences) for assistance with confocal microscopy analysis and Dr. Feng Xiong (China Zebrafish Resource Center, Institute of Hydrobiology, Chinese Academy of Sciences) for assistance with qPCR analysis. This work was supported by the Strategic Priority Research Program of the Chinese Academy of Sciences (XDB0730300), Biological Breeding-National Science and Technology Major Project (2023ZD04065), National Excellent Youth Science Fund (32322086) and the Youth Innovation Promotion Association provided funding to Shun Li. National Key Research and Development Program of China (2023YFD2400201) provided funding to Dan-Dan Chen. National Natural Science Foundation of China (32173023) provided funding to Long-Feng Lu.

## Additional information

### Funding

| Funder | Grant reference number | Author |
|---|---|---|
| Strategic Priority Research program of The Chinese Academy of Sciences | XDB0730300 | Gang Zhai<br>Shun Li |
| National Excellent Youth Science Fund | 32322086 | Shun Li |
| Youth Innovation Promotion Association of the Chinese Academy of Sciences | | Shun Li |
| National Key Research and Development Program of China | 2023YFD2400201 | Dan-Dan Chen |
| National Natural Science Foundation of China | 32173023 | Long-Feng Lu |

The funders had no role in study design, data collection and interpretation, or the decision to submit the work for publication.

### Author contributions

Long-Feng Lu, Conceptualization, Data curation, Funding acquisition, Writing – original draft; Bao-jie Cui, Sheng-Chi Shi, Validation, Investigation; Yang-Yang Wang, Na Xu, Methodology; Can Zhang, Xiao Xu, Zhuo-Cong Li, Investigation; Meng-Ze Tian, Zhen-Qi Li, Validation; Dan-Dan Chen, Supervision, Funding acquisition; Li Zhou, Gang Zhai, Zhan Yin, Writing – review and editing; Shun Li, Conceptualization, Data curation, Supervision, Funding acquisition, Writing – review and editing

### Author ORCIDs

Long-Feng Lu ⓘ https://orcid.org/0009-0002-8537-1726
Zhan Yin ⓘ https://orcid.org/0000-0002-7969-3967
Shun Li ⓘ https://orcid.org/0000-0002-3629-9900

### Ethics

The experiments involved in this study were conducted in compliance with ethical regulations. The fish experiments were carried out under the guidance of the European Union Guidelines for the Handling of Laboratory Animals (2010/63/EU) and approved by the Ethics Committee for Animal Experiments of the Institute of Hydrobiology, Chinese Academy of Sciences (CAS) (No. 2024-069).

Reviewer #1 (Public review): https://doi.org/10.7554/eLife.108048.4.sa1
Reviewer #2 (Public review): https://doi.org/10.7554/eLife.108048.4.sa2
Author response https://doi.org/10.7554/eLife.108048.4.sa3

## Additional files

### Supplementary files

Supplementary file 1. List of primer sequences used in this study.

Supplementary file 2. List of all identified genes in FHK-vs-MHK in transcriptomic analysis.

Supplementary file 3. Heatmap view of mRNA variations of SVCV-activated ISG sets in the Cyp17a2-overexpressing cells or *cyp17a2* knockdown cells.

MDAR checklist

## Data availability

RNA sequence data were deposited in GEO (accession number: GSE286486). All data generated or analyzed during this study are included in the manuscript and supporting files; source data files have been provided for all figures.

The following dataset was generated:

| Author(s) | Year | Dataset title | Dataset URL | Database and Identifier |
|---|---|---|---|---|
| Lu L, Shi S | 2025 | Male-biased gene cyp17a2 enhances virus resistance in fish by promoting STING expression and degrading viral protein | https://www.ncbi.nlm.nih.gov/geo/query/acc.cgi?acc=GSE286486 | NCBI Gene Expression Omnibus, GSE286486 |

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
