## [Editor Report · eLife Assessment]

This **valuable** study describes an interesting infection phenotype that differs between adult male and female zebrafish. The authors present data indicating that male-biased expression of Cyp17a2 appears to mediate viral infection through STING and USP8 activity regulation. Through experimentation on male fish, the authors present **solid** evidence linking this factor to direct and indirect antiviral outcomes through ubiquitination pathways. These findings raise interesting questions about immune mechanisms that underlie sex-dimorphism and the selective pressures that might shape it.

---

## [Referee Report · Reviewer #1 (Public review)]

Summary:

In this manuscript Lu & Cui et al. observe that adult male zebrafish are more resistant to infection and disease following exposure to Spring Viremia of Carp Virus (SVCV) than female fish. The authors then attempt to identify some of the molecular underpinnings of this apparent sexual dimorphism and focus their investigations on a gene called cytochrome P450, family 17, subfamily A, polypeptide 2 (cyp17a2) because it was among genes that they found to be more highly expressed in kidney tissue from males than in females. Their investigations lead them to propose a direct connection between cyp17a2 and modulation of interferon signaling as the key underlying driver of difference between male and female susceptibility to SVCV.

Strengths:

Strengths of this study include the interesting observation of a substantial difference between adult male and female zebrafish in their susceptibility to SVCV, and also the breadth of experiments that were performed linking cyp17a2 to infection phenotypes and molecularly to the stability of host and virus proteins in cell lines. The authors place the infection phenotype in an interesting and complex context of many other sexual dimorphisms in infection phenotypes in vertebrates. This study succeeds in highlighting an unexpected factor involved in antiviral immunity that will be an important subject for future investigations of infection, metabolism, and other contexts.

Weaknesses:

Weaknesses of this study include a proposed mechanism underlying the sexual dimorphism phenotype based on experimentation in only males, and widespread reliance on over-expression when investigating protein-protein interaction and localization.

---

## [Referee Report · Reviewer #2 (Public review)]

This study conducted by Lu et al. explores the molecular underpinnings of sexual dimorphism in antiviral immunity in zebrafish, with a particular emphasis on the male-biased gene cyp17a2. The authors demonstrate that male zebrafish exhibit stronger antiviral responses than females, and they identify a teleost-specific gene cyp17a2 as a key regulator of this dimorphism. Utilizing a combination of in vivo and in vitro methodologies, they demonstrate that Cyp17a2 potentiates IFN responses by stabilizing STING via K33-linked polyubiquitination and directly degrades the viral P protein via USP8-mediated deubiquitination. The work challenges conventional views of sex-based immunity and proposes a novel, hormone- and sex chromosome-independent mechanism.

Strengths:

(1) The following constitutes a novel concept, sexual dimorphism in immunity can be driven by an autosomal gene rather than sex chromosomes or hormones represents a significant advance in the field, offering a more comprehensive understanding of immune evolution.

(2) The present study provides a comprehensive molecular pathway, from gene expression to protein-protein interactions and post-translational modifications, thereby establishing a link between Cyp17a2 and both host immune enhancement (via STING) and direct antiviral activity (via viral protein degradation).

(3) In order to substantiate their claims, the authors utilize a wide range of techniques, including transcriptomics, Co-IP, ubiquitination assays, confocal microscopy, and knockout models.

(4) The utilization of a singular model is imperative. Zebrafish, which are characterized by their absence of sex chromosomes, offer a clear genetic background for the dissection of autosomal contributions to sexual dimorphism.

---

## [Author Response]

The following is the authors’ response to the previous reviews

**Public Reviews:**

**Reviewer #1 (Public review):**
Weaknesses:(1) Weaknesses of this study include a proposed mechanism underlying the sexual dimorphism phenotype based on experimentation in only males, and widespread reliance on over-expression when investigating protein-protein interaction and localization. Additionally, a minor weakness is that the text describing the identification of cyp17a2 as a candidate contains errors that are confusing.

We thank the reviewer for these insightful comments, which have helped us improve the manuscript.

(1) Experimentation in males. We focused on male zebrafish for our mechanistic studies to preclude potential confounding effects from female hormones and to directly interrogate the basis of the observed male-biased resistance. As confirmed in the manuscript (lines 151-153), both wild-type and cyp17a2⁻/⁻ males developed normal male sex organs and exhibited comparable androgen levels. This crucial control gives us confidence that the differences in antiviral immunity we observed are a direct consequence of Cyp17a2 loss-of-function, rather than secondary to developmental or hormonal abnormalities. We fully agree that elucidating the mechanism in females represents a valuable and interesting direction for future research.

(2) Over-expression studies. We acknowledge that overexpression approaches can have inherent limitations. To mitigate this and strengthen our conclusions, we complemented these experiments with loss-of-function data from both knockout zebrafish and knockdown cells, as well as validation at the endogenous level (e.g., Fig. 4J and S4C). The consistent results obtained across these diverse experimental models collectively reinforce our conclusion that Cyp17a2 interacts with and stabilizes STING.

(3) We thank the reviewer for pointing out the lack of clarity in the text regarding the selection process of Cyp17a2. We have thoroughly revised the manuscript to provide a precise and accurate description of our methodology. The relevant text is now as follows: “Differential expression analysis identified 1511 upregulated and 1117 downregulated genes (Fig. 2A and Table S2). We then focused on a subset of known or putative sexrelated genes. Among these eight candidates, cyp17a2 exhibited the most significant male-biased upregulation, a finding that was subsequently confirmed by qPCR (Fig. 2B and S1A)” (lines 142-144).

(2) Lines 139-140 describe the data for Figure 2 as deriving from "healthy hermaphroditic adult zebrafish". This appears to be a language error and should be corrected to something that specifies that the comparison made is between healthy adult male and female kidneys.

We thank the reviewer for pointing out this inaccuracy. This was a terminological error, and we have corrected the text to accurately state “transcriptome sequencing was performed on head-kidney tissues from healthy adult male and female zebrafish” (lines 139-140). We have carefully reviewed the manuscript to ensure no similar errors are present.

(3) In Figure 2A and associated text cyp17a2 is highlighted but the volcano plot does not indicate why this was an obvious choice. For example, many other genes are also highly induced in male vs female kidneys. Figure 2B and line 143 describe a subset of "eight sex-related genes" but it is not clear how these relate to Figure 2A. The narrative could be improved to clarify how cyp17a2 was selected from Figure 2A and it seems that the authors made an attempt to do this with Figure 2B but it is not clear how these are related. This is important because the available data do not rule out the possibility that other factors also mediate the sexual dimorphism they observed either in combination, in a redundant fashion, or in a more complex genetic fashion. The narrative of the text and title suggests that they consider this to be a monogenic trait but more evidence is needed.

We thank the reviewer for raising these important points. We have revised the manuscript to clarify the candidate gene selection process and to avoid any implication that the trait is monogenic.

The selection of cyp17a2 was not based solely on its position in the volcano plot (Fig. 2A), but on a multi-faceted rationale. We first prioritized genes with known or putative sex-related functions from the pool of differentially expressed genes. From this subset, cyp17a2 emerged as the lead candidate due to a combination of unique attributes, it exhibited the most significant and consistent male-biased upregulation among the validated candidates (Fig. 2B and S1A); it is a teleost-specific autosomal gene, suggesting a novel mechanism for sexual dimorphism independent of canonical sex chromosomes; and it showed conserved male-biased expression across multiple tissues (Fig. 2C and 2D). Regarding its representation in the volcano plot, cyp17a2 was included in the underlying dataset but was not explicitly labeled in the revised Figure 2A to maintain visual clarity, as the plot aimed to illustrate the global transcriptomic landscape rather than highlight individual genes.

We agree with the reviewer that other genetic factors may contribute to the observed sexual dimorphism. Accordingly, we have modified the text throughout the manuscript to remove any suggestion of a purely monogenic trait. Our functional data position cyp17a2 as a key and sufficient factor, as its knockout in males was sufficient to ablate the antiviral resistance phenotype (Fig. 2E-G), demonstrating a major, nonredundant role without precluding potential contributions from other genes.

The following specific changes have been made to the text.

(1) The title has been revised by replacing “governs” with “orchestrates.” (line 1)

(2) The abstract now states “the male-biased gene cyp17a2 as a critical mediator of this enhanced response” instead of “which are driven by the male-biased gene Cyp17a2 rather than by hormones or sex chromosomes.” (lines 33-34)

(3) The discussion now states “Our study leverages this unique context to demonstrate that enhanced antiviral immunity in males is mediated by the male-biased expression of the autosomal gene cyp17a2,” removing the comparative phrasing regarding hormones or sex chromosomes. (lines 364-366)